# Calibrating macro-scale hydrological models in poorly gauged and heavily regulated basins

Dung Trung Vu[1], Thanh Duc Dang[2], Francesca Pianosi[3], and Stefano Galelli[1,4]

[1]Pillar of Engineering Systems and Design, Singapore University of Technology and Design, Singapore
[2]Department of Civil and Environmental Engineering, University of South Florida, Tampa, FL, USA
[3]School of Civil, Aerospace and Design Engineering, University of Bristol, Bristol, UK
[4]School of Civil and Environmental Engineering, Cornell University, Ithaca, NY, USA

**Correspondence:** Dung Trung Vu (trungdung_vu@mymail.sutd.edu.sg)

**Abstract.** The calibration of macro-scale hydrological models is often challenged by the lack of adequate observations of river discharge and infrastructure operations. This modelling backdrop creates a number of potential pitfalls for model calibration, potentially affecting the reliability of hydrological models. Here, we introduce a novel numerical framework conceived to explore and overcome these pitfalls. Our framework consists of VIC-Res (a macro-scale model setup for the Upper Mekong River Basin) and a hydraulic model used to infer discharge time series from satellite data. Using these two models and Global Sensitivity Analysis, we show the existence of a strong relationship between the parameterization of the hydraulic model and the performance of VIC-Res—a co-dependence that emerges for a variety of performance metrics we considered. Using the results provided by the sensitivity analysis, we propose an approach for breaking this co-dependence and informing the hydrological model calibration, which we finally carry out with the aid of a multi-objective optimization algorithm. The approach used in this study could integrate multiple remotely sensed observations and is transferable to other poorly gauged and heavily regulated river basins.

## 1 Introduction

The past few years have witnessed an increase in the implementation of hydrological models to extensive domains, from large basins to continental or even global scale (Döll et al., 2008; Haddeland et al., 2014; Nazemi and Wheater, 2015a, b; Bierkens, 2015) for a variety of downstream applications, such as quantifying the potential impact of climate change on water resources (van Vliet et al., 2016), characterizing the relationship between climate, water, and energy (Chowdhury et al., 2021), or predicting extreme events over multiple time scales (Vegad and Mishra, 2022). Such macro-scale hydrological models are rarely calibrated and, when they are, calibration is typically limited to a portion of the modelled domain (Bierkens, 2015). This is due to the high computational cost of calibration at large scales but also, and more importantly, to the lack of long and reliable time series of in situ river discharge observations in many regions of the world (Hrachowitz et al., 2013). In poorly

gauged basins, model calibration is sometimes carried out by leveraging the few discharge data that are available (e.g., Shin et al. (2020); Galelli et al. (2022); Chuphal and Mishra (2023)). Naturally, doing so potentially leads to inadequate model calibration for the ungauged regions of the domain. An additional problem is the lack of information and data on the operations of hydraulic infrastructures; a matter that we have only recently started to address (Vu et al., 2022; Steyaert et al., 2022). This is important because hydraulic infrastructures, such as dams, are ubiquitous and heavily affect hydrological processes (Haddeland et al., 2006; Grill et al., 2019) and therefore, if not properly accounted for, the results of model calibration. For instance, Dang et al. (2020a) showed that a macro-scale hydrological model ignoring dams presence can be calibrated to attain the same level of fit-to-data as a model that explicitly represent dams; however, such performance is attained through "optimally calibrated" soil parameters that are unrealistic, and are selected to compensate for the structural error of neglecting dams, ultimately biasing the representation of both natural and human-impacted hydrological processes. Importantly, both problems highlighted here are exacerbated in transboundary river basins, where access to data is particularly difficult.

Some studies have explicitly dealt with the lack of in-situ discharge time series by inferring discharge from satellite data. As shown in Figure 1, these studies can be categorized into two groups. One approach (panel (a)) first develops a hydraulic model for estimating river discharge from remote-sensed water level and/or river width, and then uses these estimates to carry out the calibration of the hydrological model (Khan et al., 2012; Tarpanelli et al., 2022). This approach still partially relies on in-situ data. For example, Xiong et al. (2021) converted remote-sensed water level to river discharge via a rating curve—the relationship between river discharge and water level—for calibrating their GR6J hydrological model. The rating curve was developed based on Manning's equation, using surveyed river cross-sections and a few pairs of in-situ discharge and remote-send water level data for validation. When these data are not available, another possible approach (panel (b)) is to calibrate both models concurrently (e.g., Liu et al. (2015); Sun et al. (2018); Huang et al. (2020)). Here, a potential pitfall stands in the fact that estimation errors in the hydraulic model (discharge estimation) may be compensated by introducing parameter biases in the hydrological model, and vice versa (Lima et al., 2019). In other words, simultaneous calibration of the hydraulic and hydrological models may yield biased parameters, ultimately compromising the realism and reliability of the calibrated models.

Considering the increasing number of remotely sensed hydrological data that have become available over the last decades (Birkinshaw et al., 2010; Papa et al., 2012; Biancamaria et al., 2016), and that in many regions of the world these satellite products are the only means to estimate river discharges, the question arises on how to best use such remotely sensed data to support model calibration. Hence, the overarching question that this paper addresses is: to what extent is it possible and helpful to calibrate a macro-scale hydrological model in ungauged catchments using remotely sensed data? How do we deal with potential interactions between parameters used in data pre-processing (i.e., from remotely sensed data to reconstructed discharge data) and parameters of the hydrological models when doing model calibration? Can we reduce the uncertainty from such interactions in model calibration results? We answer these questions for an implementation of the VIC-Res hydrological model for the Upper Mekong River Basin (Dang et al., 2020a), an area characterized by the unavailability of discharge observations as well as major hydrological alterations caused by dam development (Hecht et al., 2019). To generate discharge time series for the calibration of VIC-Res, we use satellite altimetry data and a hydraulic model (based on the Manning's equation) that is

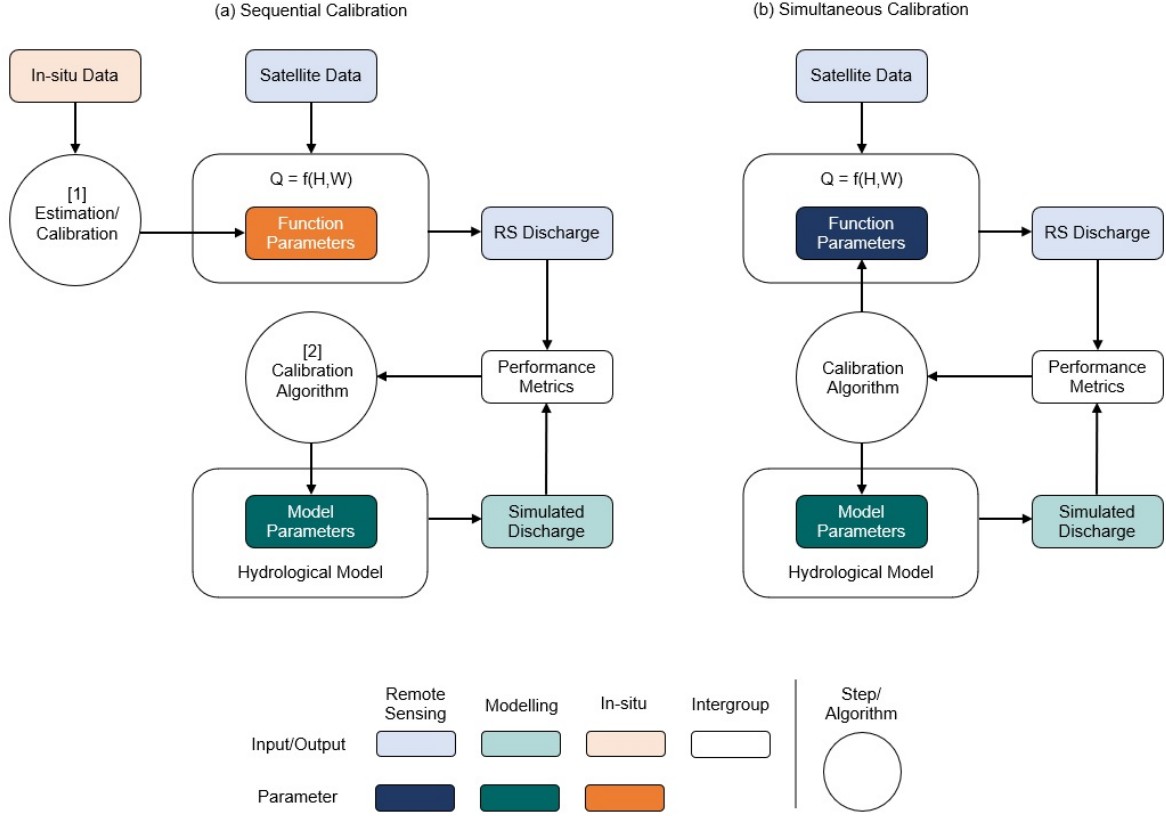

**Figure 1.** Two approaches to the calibration of macro-scale hydrological models with discharge data retrieved from satellite data. (a) With the sequential calibration, the discharge data are first estimated using a hydraulic model linking water level ($H$) and/or river width ($W$) to discharge ($Q$), and then used to calibrate the hydrological model. (b) With the second approach, both models are calibrated simultaneously.

also identified from satellite data. In our framework, we first use Global Sensitivity Analysis to demonstrate the existence of a pronounced co-dependence between the parameterization of the hydraulic model and the modelling accuracy of VIC-Res. To break this co-dependence, we leverage the results of the sensitivity analysis to constrain the parameterization of the hydraulic model and thus safely inform the calibration of VIC-Res, which is ultimately carried out using a multi-objective optimization approach.

## 2 Study site, model domain, and gauging stations

In this section, we provide information on our study site, the spatial domain of the hydrological model, and the availability of observed and remote-sensed discharge data.

## 2.1 The Lancang-Mekong River Basin

Spanning an area of about 795,000 km$^2$, the Mekong River Basin is the largest transboundary basin in Southeast Asia. The river is 4,350 km long and stretches in a northwest-southeast direction from the Tibetan Plateau (approximately 5,200 m a.s.l.) to the East Vietnam Sea (Figure 2a). The basin can be roughly divided into two parts, namely the Upper Mekong (also known as the Lancang, in China) and the Lower Mekong, which is shared by five countries (Myanmar, Thailand, Laos, Cambodia, and Vietnam).

The Lancang accounts for 45% of the river length, 21% of the catchment area, and 16% of the annual discharge of the entire Mekong (MRC, 2009). The complex topography of the Lancang Basin (high mountains and low valleys) contributes to the uneven spatial distribution of precipitation, which ranges from 600 mm/year in the Tibetan Plateau to 1700 mm/year in the mountains of Yunnan. Meanwhile, the monsoonal climate causes an uneven temporal distribution of precipitation, with 70-80% of precipitation arriving in the wet season (June to November) (Yun et al., 2020).

Because of the advantageous topography and abundant water availability, the Lancang River Basin has become a hotspot for hydropower development. Indeed, the Lancang dam system—developed during the past three decades—consists of more than 35 hydropower dams (WLE Mekong, n.d.), including 10 large dams on the main stem with a volume larger than 100 MCM (Million Cubic Meters) each (see their location in Figure 2a and specifications in Table S1). The system has a total capacity of more than 42,000 MCM and can control up to 55% of the annual flow to Northern Thailand and Laos. The Lancang River Basin is an excellent example of a transboundary and heavily regulated river with limited information on dam operations: initiatives on the sharing of year-round water data are still in their infancy (Johnson, 2020), so the only data available to the public are those retrieved from satellite data (e.g., Bonnema and Hossain (2017); Biswas et al. (2021); Vu et al. (2022)). Time series of river discharge measured within China's political boundaries are not available.

## 2.2 Model domain and study period

The spatial domain of our hydrological model is the light green area illustrated in Figure 2. This domain corresponds to the Lancang basin (namely the area falling within China's political boundaries), plus an additional area spanning across Myanmar, Thailand, and Laos. Note that the domain of hydrological models focusing on the Lancang is typically 'closed' at Chiang Saen (e.g., Dang et al. (2020a)), where the first gauging station with publicly-available data is located. Here, we slightly extend the domain so as to account for the location of a virtual gauging station (see Section 2.3). The simulation period goes from 2009 to 2018 and thus comprises the main development of the Lancang reservoir system, including the filling period of the two largest reservoirs, Xiaowan and Nuozhadu, which account for ∼85% of the total system's storage (Vu et al., 2022). Another reason for the choice of this study period is to make it compatible with the temporal coverage of altimetry data at our virtual station, which we describe next.

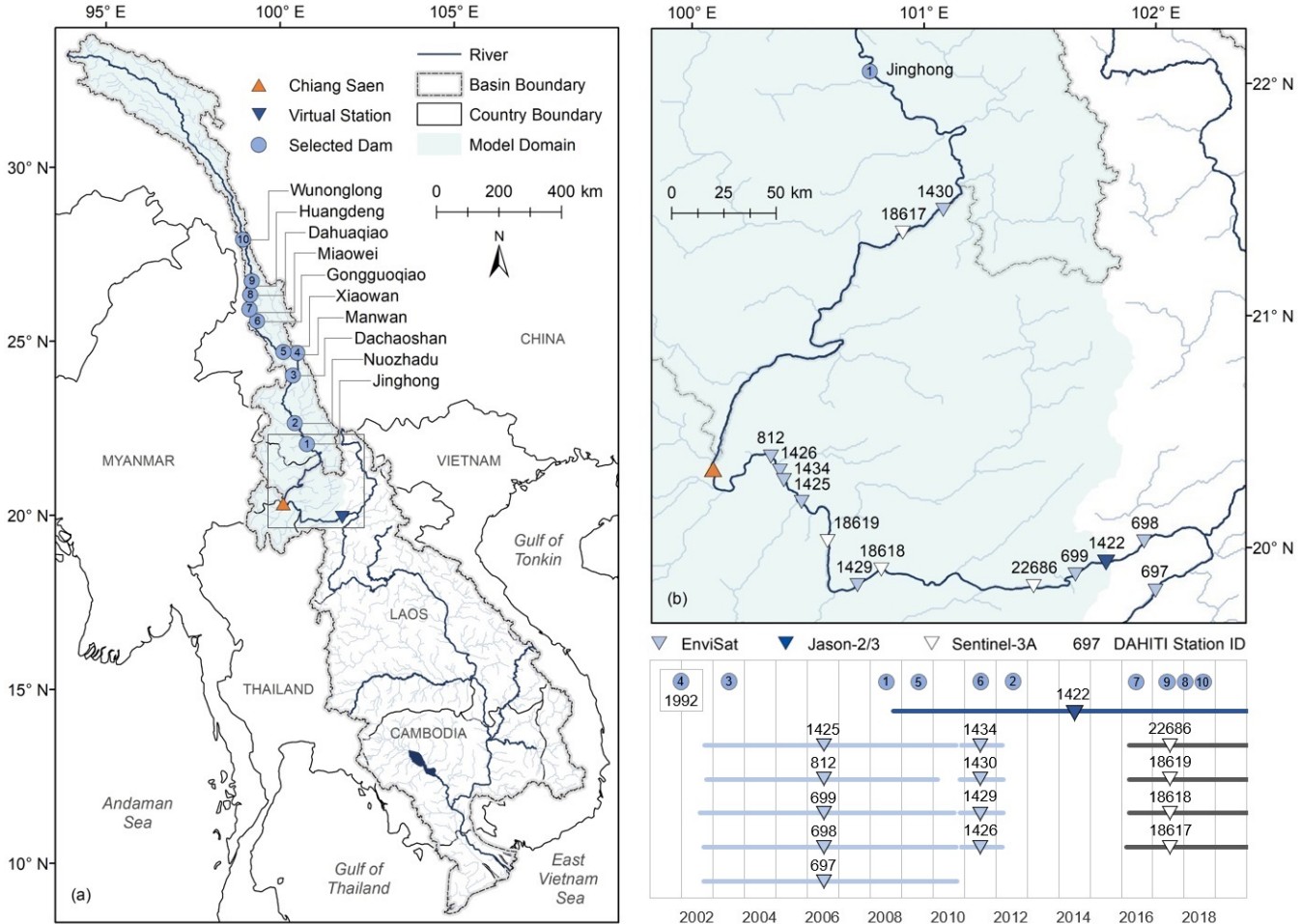

**Figure 2.** Panel (a) shows the Mekong River Basin and its upper portion—the Lancang River Basin. We illustrate, in this panel, the location of the gauging station (Chiang Saen), virtual gauging station, and ten large hydropower dams on the main stem of the Lancang with a volume larger than 100 million m$^3$ each, all included in the hydrological model. All dams are operational as of December 2020. The light green area is the spatial domain of the hydrological model. Panel (b) illustrates the locations around Chiang Saen, in which altimetry water level data are available. The data are collected by multiple satellites—namely EnviSat (light blue triangle), Jason-2/3 (dark blue triangle), and Sentinel-3A (white triangle)—and are processed by DAHITI. The number above each triangle corresponds to the station ID in DAHITI. The lower part of panel (b) illustrates the commission year of each dam and the temporal coverage of altimetry data in each location, constrained by the operational period of the satellites. The location 1422 is chosen as our virtual station because of the temporal coverage and resolution of altimetry water level data at this location as well as its suitability to apply the methods for constructing river cross-section and rating curve (see Section 3.2).

## 2.3 Gauging stations

As mentioned above, the first gauging station with publicly-available data is Chiang Saen, located in Northern Thailand, about 350 km from Jinghong dam (Figure 2). Daily water level and discharge at the station have been collected since 1990 by the Mekong River Commission (MCR) and are available on its online data portal (https://portal.mrcmekong.org/). Since we developed a methodology for calibrating models in ungauged river basins, these data are used only for model validation.

To infer the discharge time series needed for model calibration, we sought for locations around Chiang Saen where altimetry water level data are available (Figure 2b). From these data, one can try to infer the river discharge. These data are collected by multiple satellites (i.e., EnviSat, Jason-2/3, and Sentinel-3A) and are available in the Database for Hydrological Time Series of Inland Waters (DAHITI, https://dahiti.dgfi.tum.de/). In this study, we choose the location 1422 (Jason-2/3)—about 280 km downstream of Chiang Saen—as our *virtual* gauging station (virtual station hereafter). This is because of two main reasons. First, the temporal coverage of data at the chosen location covers the commission year of the majority of the dams, including the two largest reservoirs, Xiaowan and Nuozhadu (see the lower part of Figure 2b). Second, the temporal resolution of Jason-2/3 (10 days) is finer than the one of EnviSat (35 days) and Sentinel-3A (27 days). It is also worth noting that another database, HydroWeb (https://hydroweb.theia-land.fr/), provides (Sentinel-3A/B) altimetry water level data for a number of locations in our study site. However, these data have the same temporal resolution and coverage as the Sentinel-3A data provided by DAHITI, which makes them unsuitable for our study. Moreover, the methods used to construct river cross-section and rating curve at the virtual station work best for location 1422 (see Section 3.2).

## 3 Methodology

The numerical framework developed for our study consists of two main modelling components, illustrated in Figure 3. We model the hydrological processes within the Lancang Basin with VIC-Res, whose routing module includes an explicit representation of reservoir operations (Section 3.1). The discharge data at the virtual station used to calibrate VIC-Res are generated by a simple hydraulic model, namely a rating curve based on the Manning's equation (Section 3.2). In our approach, we first use Global Sensitivity Analysis to explore the relationship between the parameterization of the rating curve and the performance (fit-to-data) that can be achieved through calibration of VIC-Res (Section 3.3). Then, we use the knowledge gained through this sensitivity analysis to select the parameterisation of the rating curve and proceed with the calibration and validation of VIC-Res.

### 3.1 Modelling hydrological processes and reservoir operations

#### 3.1.1 Hydrological model

The hydrological model used in this study is VIC-Res (Dang et al., 2020a), a novel variant of VIC, which is a macro-scale, semi-distributed hydrological model developed by the University of Washington (Liang et al., 2014). Both VIC and VIC-Res consist of two modules, namely a rainfall-runoff and a routing module (Figure 3). In the rainfall-runoff module, the study

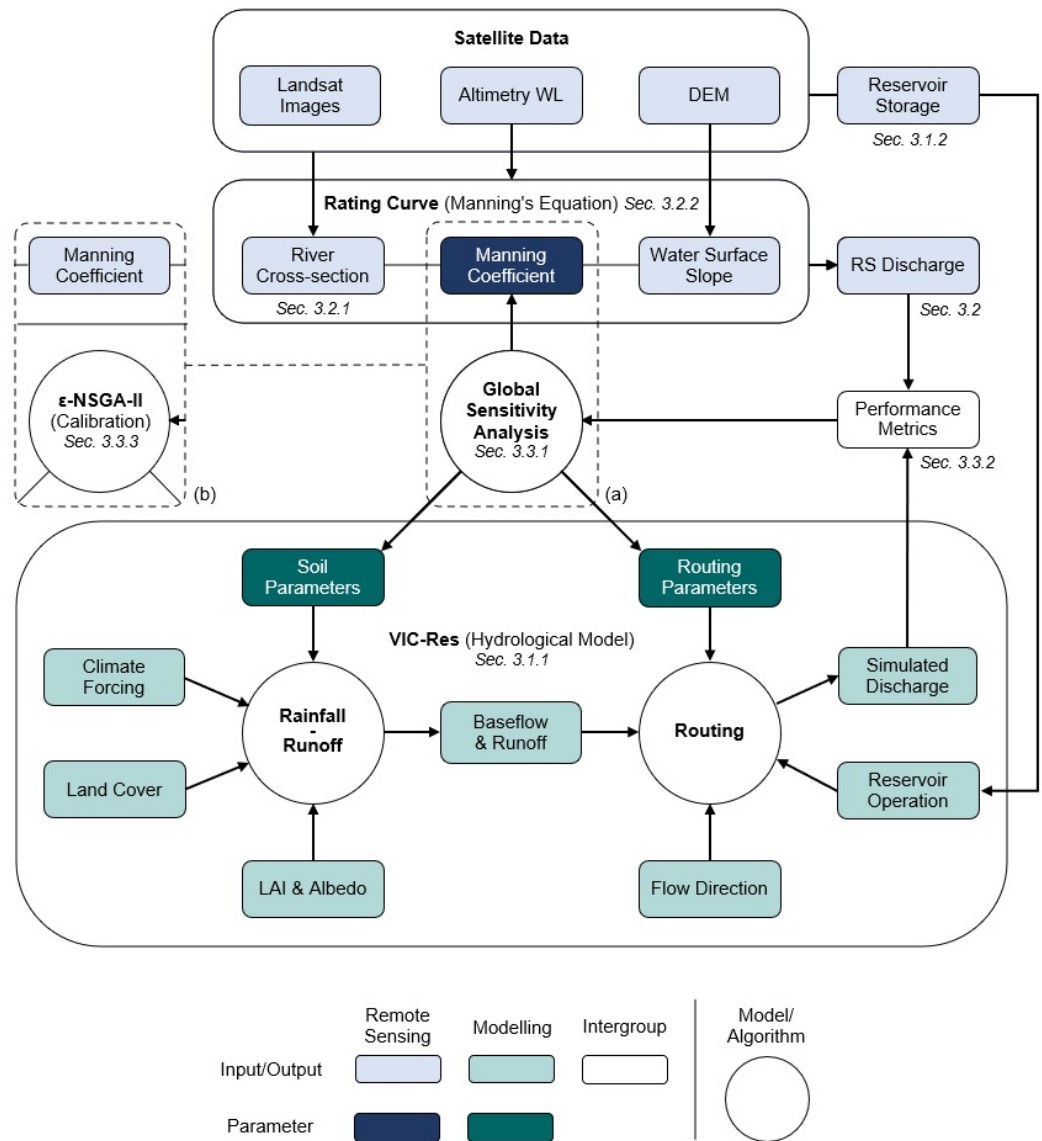

**Figure 3.** Flowchart illustrating our numerical framework. The VIC-Res model (green boxes) includes a rainfall-runoff and a routing module. The latter explicitly simulates reservoir operations using data retrieved from satellite observations. The discharge data used to calibrate VIC-Res are estimated from altimetry water levels through a rating curve, which is based on Manning's equation and developed using multiple satellite data (Landsat images, altimetry water level, and DEM). All remote sensing items are represented by blue boxes. The relationship between the parameterization of the Manning's equation (dark blue box) and the performance of VIC-Res is assessed and quantified via Global Sensitivity Analysis (a). Based on the results of the sensitivity analysis, we then set a value of the Manning's coefficient and calibrate the parameters of VIC-Res using the $\varepsilon$-NSGA-II algorithm (b).

region is divided into computational cells with a customizable cell size (0.0625 degrees in this study). For each cell, the key hydrological processes (evapotranspiration, infiltration, baseflow, and runoff) are calculated as a function of various inputs, including climate forcing (e.g., precipitation, temperature, wind speed), land cover, Leaf Area Index, and albedo. In the routing module, simulated baseflow and runoff produced by the first module are routed throughout the river network, with the routing
process modelled by the linearized Saint-Venant equation (Lohmann et al., 1996, 1998).

  Improving on the VIC model, VIC-Res includes an explicit representation of water reservoir operations. For each reservoir in the study region, the model solves the storage mass balance and calculates the reservoir release. Specifically, we leverage information on modeled inflow and estimated storage (see Section 3.1.2). These two variables are combined with information on evaporation (simulated using the estimated water surface area and evaporation rates calculated with the Penman equation) to
135 invert the mass balance equation, yielding the reservoir release. Additional details on VIC-Res, including alternative approaches to reservoir operations, are described in Dang et al. (2020b).

  In our VIC-Res model, we calibrate 7 soil parameters and 2 routing parameters (see Table 1). The soil parameters controlling the rainfall-runoff process are $b$, $D_{max}$, $D_S$, $W_S$, $c$, $d_1$, and $d_2$. To be more specific, the parameter $b$ is the VIC curve parameter, which determines the infiltration capacity and surface runoff amount generated by each cell (Ren-Jun, 1992; To-
140 dini, 1996). In particular, higher values of $b$ produce less infiltration and more surface runoff. $D_{max}$, $D_S$, $W_S$, and $c$ are the baseflow parameters, which influence the shape of the baseflow curve (Franchini and Pacciani, 1991). Specifically, $D_{max}$ is the maximum velocity of baseflow, $D_S$ is the fraction of $D_{max}$ at which non-linear baseflow begins, while $W_S$ is the fraction of maximum soil moisture at which non-linear baseflow begins. The parameter $c$ is the exponent used in the baseflow curve. $d_1$ and $d_2$ are the thickness of the two soil layers. Thicker layers increase the water storage capacity, and hence increase the
145 evaporation losses. Thicker soil layers also delay the seasonal peak flow. The routing parameters are flow velocity ($v$) and flow diffusion ($d_f$).

  The data used in our VIC-Res model consist of climate forcing data, land use and cover, Leaf Area Index (LAI), albedo, flow direction, and time series of reservoir storage volume. Climate forcing data include daily precipitation data retrieved from the CHIRPS-2.0 dataset, daily maximum and minimum temperature, and wind speed (retrieved from the ERA5 dataset).
We collect land use and cover data from the Global Land Cover Characterization (GLCC) dataset, and soil data from the Harmonized World Soil Database (HWSD). Monthly LAI and albedo are derived from the Terra MODIS satellite images, while the flow direction is calculated from the SRTM-DEM data. The monthly time series of reservoir storage volume are reconstructed from satellite data, as explained below.

  We finally note that the choice of the cell size could affect the rainfall-runoff and routing estimations, and thus impact
model calibration and simulated discharge (Egüen et al., 2012). Since the issue applies to any modelling exercise, not only to those relying on remotely sensed data like this study, we do not carry out an analysis of the impact of cell size on model performance. Instead, we choose a cell size (i.e., 0.0625°) that falls in between what is currently being adopted for the existing distributed models for the Mekong region. For example, Costa-Cabral et al. (2007) and Tatsumi and Yamashiki (2015) adopted a resolution of 1 / 12°and 0.25°, while Du et al. (2020) and Bonnema and Hossain (2017) used a resolution of 90m / 900m and
0.01°, respectively.

**Table 1.** Soil parameters controlling the rainfall-runoff process and routing parameters in VIC-Res. The last column shows the range of each parameter considered in this study, also adopted in previous studies (e.g., Dan et al. (2012); Park and Markus (2014); Xue et al. (2015); Wi et al. (2017)).

| | Parameter | Unit | Description | Range |
|---|---|---|---|---|
| | $b$ | - | Variable Infiltration Capacity curve parameter | (0, 0.9] |
| | $D_{max}$ | mm/d | Maximum velocity of baseflow | (0, 30] |
| | $D_S$ | - | Fraction of $D_{max}$ where non-linear baseflow occurs | (0, 1) |
| Soil | $W_S$ | - | Fraction of maximum soil moisture where non-linear baseflow occurs | (0, 1) |
| | $c$ | - | Exponent used in baseflow curve | [1, 3] |
| | $d_1$ | m | Thickness of the upper soil layer | [0.05, 0.25] |
| | $d_2$ | m | Thickness of the lower soil layer | [0.3, 1.5] |
| Routing | $v$ | m/s | Flow velocity | [0.5, 5] |
| | $d_f$ | m$^2$/s | Flow diffusion | [200, 4000] |

### 3.1.2 Reservoir operations

To capture the actual operations of reservoirs, we use monthly time series of reservoir storage volume reconstructed from satellite data by Vu et al. (2022). Specifically, the time series of reservoir storage volume are obtained from Landsat images (Landsat 5 available from 1984 to 2013, Landsat 7 from 1999 to 2022, and Landsat 8 from 2013 to present) and a digital elevation model (SRTM-DEM). The time series are created through three steps. First, the relationship between water surface area and storage volume (the area–storage curve) for each reservoir is calculated from DEM data. Then, the reservoir water surface area is estimated from Landsat images by a water surface area estimation algorithm that removes the effects of clouds and other disturbances (Gao et al., 2012; Zhang et al., 2014). Finally, the storage volume is inferred from the water surface area through the area–storage curve. The results obtained from Landsat images are validated with altimetry water levels (Jason 2 available from 2008 to 2016, Jason 3 from 2016 to present, and Sentinel 3 from 2016 to present) for the reservoirs where altimetry water levels are available. Since the VIC-Res model adopts a daily simulation time step, the monthly time series of reservoir storage volume is interpolated to daily values. Although using interpolated values (monthly to daily) is not ideal, it is reasonable to do so if one considers the specific characteristics of the reservoir system. In particular, Xiaowan and Nuozhadu are the two largest reservoirs: they have a massive capacity ($\sim$36 km$^3$) and account for about 85 % of the total system's storage. Because of their size, their role is not to follow inter and intra-daily electricity demand variability, but rather to ensure a stable supply of power and to minimize the variability in the production of the other dams composing the hydropower system. This goal is reflected by their operating patterns. In the wet season (June-November), Xiaowan and Nuozhadu reservoirs gradually

store water until reaching their maximum operational level (and release extra water if necessary). The other reservoirs run at their normal operational level (full capacity for power generation). In the dry season (December-May), Xiaowan and Nuozhadu gradually release water to the downstream reservoirs to ensure that the other reservoirs can run at their normal operational level (International Rivers, 2014). Therefore, it is fair to state that Xiaowan and Nuozhadu are characterized by slow-varying dynamics. Additionally, the analysis carried out in Vu et al. (2022) shows a strong similarity between the monthly storage of Xiaowan and Nuozhadu derived from Landsat images and the storage derived from Jason altimetry data (10-day temporal resolution) and Sentinel-1/2 images (6-day temporal resolution). Because of the spatial and temporal coverages of those data, we use the result derived from Landsat images for this study.

## 3.2  Inferring discharge data

To handle the lack of discharge data for model calibration, we again resort to satellite data. Specifically, we convert altimetry water levels (Jason 2/3) to discharge through a rating curve specified for the location of the virtual station (see Figure 3). The rating curve (i.e., Manning's equation) is identified based on the information on river cross-section and water surface slope at the virtual station, which are also derived from satellite data.

### 3.2.1  River cross-section

We construct the river cross-section at the virtual station by using multiple satellite products (see Figure S1a). First, we use a digital elevation model (SRTM-DEM), which has a spatial resolution of 30 m, to obtain the portion of the cross-section above the water level at the observation time of the SRTM satellite (February 2000). To extend the information available to estimate the river cross-section, we then pair data on river widths at the virtual station with the corresponding water levels (temporal nearest observations of two satellites that provide river widths and water levels) (Bose et al., 2021). River widths are estimated from the water pixels—classified from Landsat images based on Normalized Difference Water Index (NDWI)—along the river cross-section. NDWI is calculated using the Green and Near-infrared bands of Landsat images (NDWI = (Green band - Near-infrared band)/(Green band + Near-infrared band)) (Zhai et al., 2015). All these bands have a spatial resolution of 30 m. Meanwhile, the water level data are processed from Jason-2/3 altimetry satellite data provided by DAHITI. Finally, for each river bank, we use a regression model (sixth-degree polynomial), which is the best fit to the data points obtained from the two first steps. The two models help us extrapolate the portion of the river cross-section under the lowest water level observed by the satellites. It is worth noting that the approach works best for river banks in natural conditions, where it is possible to infer the relation between river widths and water levels. It would be challenging to apply this approach at Chiang Saen, for example, where the river banks have been engineered.

### 3.2.2  Rating curve

We construct the rating curve at the virtual station with the Manning's equation (Equation 1):

$$Q = \frac{A^{5/3} S^{1/2}}{P^{2/3} n},$$ (1)

where $Q$, $A$, and $P$ are discharge, river cross-section area, and wet perimeter corresponding to the water depth $D$ (see Figure
S1b). As explained next, $A$ and $P$ are calculated from the river cross-section for different values of water depth $D$. $S$ is the
hydraulic slope, estimated from DEM data (which reflects the water surface slope at the observation time). $n$ is the Manning's
coefficient (riverbed roughness). Following Chow (1959) and Engineering ToolBox (2004), we assume that it ranges from 0.03
to 0.06.

The rating curve is constructed in two steps. First, we use Equation 1 to estimate the discharge corresponding to each
215 water depth with regular intervals of one meter (e.g., 0, 1, 2 m). After this step, we have at hand a number of data points, each
containing a value of water depth and its corresponding discharge. Then, we fit the data points by a power curve. This translates
into our rating curve. Note that when converting altimetry water level to discharge using the rating curve, we convert altimetry
water level to water depth by deducting the river bed elevation (Figure S1b). It is worth noting that this approach, based on the
Manning's equation, works best for straight river segments with limited discharge variations due to tributaries and distributaries
nearby (Przedwojski et al., 1995). This condition and the condition for constructing river cross-section mentioned above make
location 1422 the most suitable location for our virtual station, despite the fact that there are other locations closer to Chiang
Sean (e.g., the location 812), which could provide a better validation using observed data at Chiang Saen.

### 3.3 Sensitivity analysis and model calibration

#### 3.3.1 Sensitivity analysis

We carry out a Global Sensitivity Analysis (Pianosi et al., 2016) to study the relationship between the performance of VIC-Res
and the parameterization of the rating curve. We investigate a total of 10 model parameters, including 7 soil parameters of the
rainfall-runoff module, 2 parameters of the routing module, and the Manning's coefficient appearing in the rating curve. We
use Latin Hypercube Sampling to create 1,000 samples in the 10-dimensional parameter space defined by the ranges given in
Section 3.1.1 and 3.2.2. For each parameter sample, we run a simulation over the period 2009–2018 (after a warm-up period
from 2005 to 2008), and reconstruct discharge data for the same period with the rating curve. We then compare reconstructed
and simulated discharges through four performance metrics, which are described in the next subsection. Having built this input
(parameters) and output (performance metrics) dataset, we analyse the co-dependence between the performance of VIC-Res
and the Manning's coefficient. We also identify the parameter samples that map into the top 25% values of each performance
metric and analyze if, and how, such constraining on performances maps back into a constraining of the parameter values.
The simulation experiment is run on an Intel (R) Xeon (R) W-2175 CPU 2.50 GHz with 128 GB RAM running Linux Ubuntu
18.04. The total running time is about 200 hours.

### 3.3.2 Performance metrics

The performance metrics are calculated by comparing the simulated (by VIC-Res) and remote-sensed discharge at the virtual station. Because the temporal resolution of remote-sensed discharge is defined by the revisit time of altimetry satellite (approximately 10 days for Jason2/3), we calculate the performance metrics using the data of all days in which altimetry water levels are available. Among the several metrics available in literature (Dawson et al., 2010), we chose four metrics that explicitly capture different aspects of modelling accuracy. These are the Nash–Sutcliffe Efficiency (NSE), Transformed Root Mean Square Error (TRMSE), Mean Squared Derivative Error (MSDE), and Runoff Coefficient Error (ROCE). NSE and TRMSE assess the model performance on high and low flows, respectively, while MSDE accounts for the shape of the hydrograph timing errors, and noisy signals. Finally, ROCE assesses the overall water balance (Reed et al., 2013). The metrics are defined as follows:

$$NSE = 1 - \frac{\sum_{t=1}^{n}\left(Q_{Sim,t} - Q_{RS,t}\right)^2}{\sum_{t=1}^{n}\left(Q_{RS,t}^{t} - \overline{Q_{RS}}\right)^2}, \tag{2}$$

where $n$ is the number of satellite altimetry water level observations, $Q_{Sim,t}$ and $Q_{RS,t}$ are the simulated and remote-sensed discharge at the virtual station (for the observation number $t$), and $\overline{Q_{RS}}$ is the mean of the remote-sensed discharge.

$$TRMSE = \sqrt{\frac{1}{n}\sum_{t=1}^{n}(z_{Sim,t} - z_{RS,t})^2}, \tag{3}$$

where $z_{sim,t}$ and $z_{RS,t}$ represent the value of the simulated and remote-sensed discharge at the virtual station (for the observation number $t$), both transformed by the expression $z = \frac{(1+Q)^\lambda - 1}{\lambda}, (\lambda = 0.3)$. In other words, $\lambda$ scales down the values of the discharge, thus emphasizing the errors on low flows.

$$MSDE = \frac{1}{n-1}\sum_{t=1}^{n}\left((Q_{RS,t} - Q_{RS,t-1}) - (Q_{Sim,t} - Q_{Sim,t-1})\right)^2, \tag{4}$$

$$ROCE = abs\left(\frac{\overline{Q_{Sim}}}{\overline{P}} - \frac{\overline{Q_{RS}}}{\overline{P}}\right), \tag{5}$$

where $\overline{Q_{Sim}}$ is the mean of the simulated discharge at the virtual station, and $\overline{P}$ is the mean annual rainfall.

### 3.3.3 Model calibration

As we shall see, the global sensitivity analysis helps us understand the relationship between the performance of VIC-Res and the parameterization of the rating curve. Moreover, by identifying the parameter samples that map into high values of the performance metrics (here the top 25%), the analysis helps us narrow down the range of variability of (at least some of) the model parameters. However, one may still want to complete the model calibration by further seeking for combinations of the

VIC-Res parameters that optimize the performance metrics. To this purpose, we couple VIC-Res with $\varepsilon$-NSGA-II, a multi-objective evolutionary algorithm widely used for hydrological modelling applications (Reed et al., 2013; Dang et al., 2020a). Here, the decision variables are the nine parameters of VIC-Res, while the objective function is a vector consisting of the four metrics described in Section 3.3.2. Similarly to the sensitivity analysis, all metrics are calculated via simulation over the period 2009–2018, with a spin-up period going from 2005 to 2008. The $\varepsilon$-NSGA-II is set up with $\varepsilon = 0.001$, an initial population size of 10, and a number of function evaluations equal to 100. All performance metrics are normalized between 0 and 1. The calibration exercise is carried out on ten independent trials, with the best (Pareto-efficient) parameter combinations selected across the ten calibration exercises. The total run time is about 210 hours (using the same computational infrastructure adopted for the sensitivity analysis).

## 4  Results

Here, we move across three steps. First, we illustrate the results leading to the estimation of a discharge time series at the virtual station, including the identification of the river cross-section and rating curve (Section 4.1). Then, we use sensitivity analysis to show that there exists a co-dependence between the Manning's coefficient and the performance of VIC-Res, and we propose an approach to overcome this potential issue (Section 4.2). We finally calibrate VIC-Res and validate its performance using observed discharge data at Chiang Sean (Section 4.3).

### 4.1  Estimation of the remote-sensed discharge at the virtual station

#### 4.1.1  River cross-section

Figure 4a shows the river cross-section at the virtual station, constructed through the use of multiple satellite data. Specifically, each dark blue bar represents a 30 m cell of the SRTM-DEM lying along the river cross-section. These bars are connected by a series of segments representing an estimate of the cross-section above the water surface at the observation time of the SRTM satellite. That specific water surface is depicted by the horizontal dark blue line at the elevation of 293 m. The light blue lines indicate the river widths derived from 19 Landsat-5 images and water levels obtained from Jason-2/3. Additional information about these images, water levels, and corresponding collection dates are reported in Table S2. Finally, the dotted blue line represents the cross-section below the lowest observed water level. This line is created via extrapolation by two regression models (sixth-degree polynomial), which are fitted to the observations retrieved from DEM, Landsat-5, and Jason-2/3 (11 and 14 data points for the left and right banks, respectively). We also explore four alternative cross-sections, created by moving the one at the location of the virtual station 30 and 60 m (1 and 2 cells) both upstream and downstream, with the assumption that water levels at the alternative cross-sections are the same as the ones at the virtual station (water surface slope around the virtual station estimated from DEM is about $0.00015 \approx 8.8$ mm/60 m). The alternative cross-sections are well in agreement with the cross-section at virtual station (see Figure S2). Specifically, riverbed elevations are 277.2, 275.6, 276, 274.5, and 274.3 m a.s.l. (from upstream to downstream).

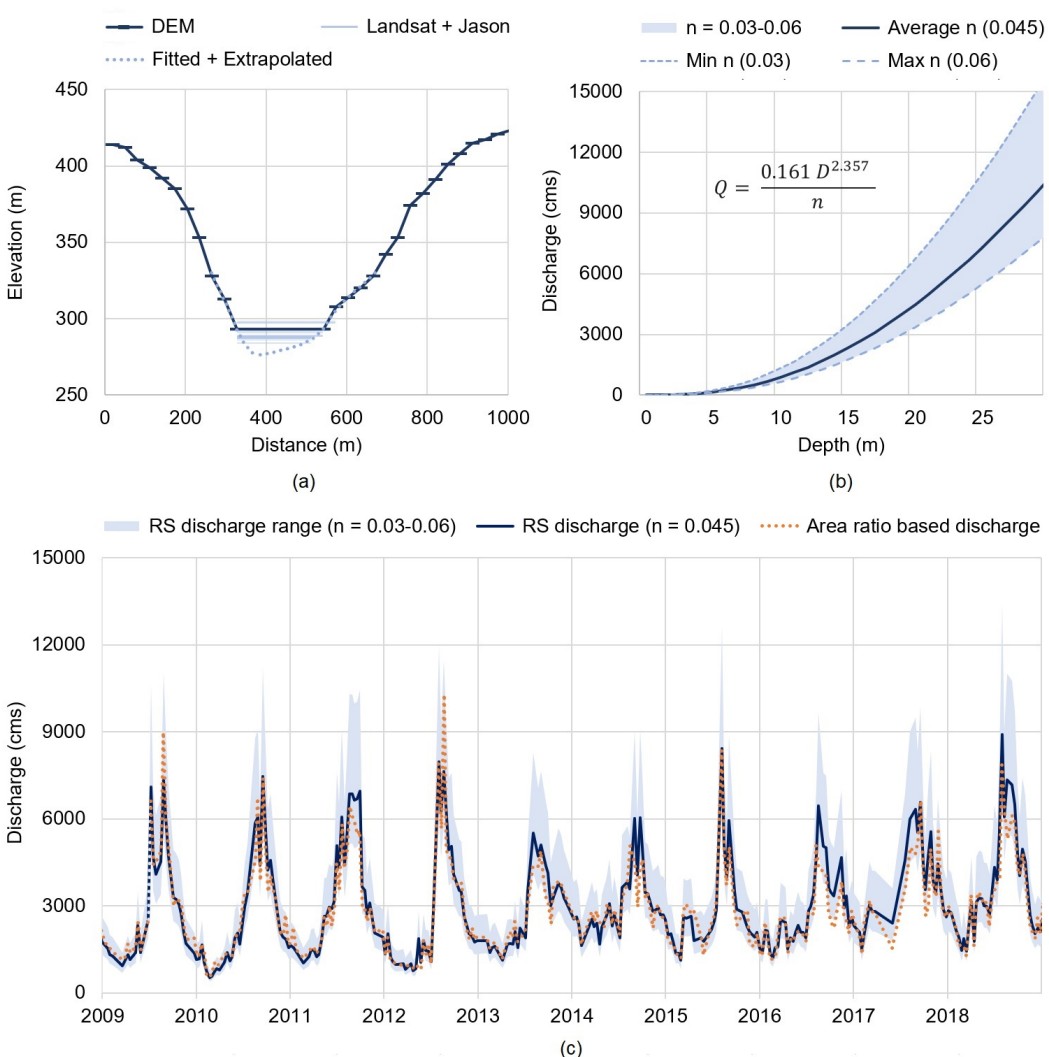

**Figure 4.** (a) River cross-section at the virtual station constructed from multiple satellite data. The dark blue line is obtained from SRTM-DEM, while the light blue lines are retrieved by paring Landsat-derived river widths with Jason altimetry water levels. The dotted blue line is created by using two regression models, which are first fitted to the right and left banks and then extrapolated to the portion below the lowest observed water level. (b) Range of variability of the rating curve (at the virtual station) for values of $n$ ranging from 0.003 to 0.006 (light blue band). In this plot, we also illustrate three rating curves corresponding to specific values of $n$: minimum (dotted blue line), average (dark blue line), and maximum (dashed blue line). (c) Remote-sensed (RS) discharge at the virtual station. The light blue band represents the range of variability, with $n$ varying from 0.03 to 0.06. The dark blue line is the estimated discharge with the average value of $n$ (0.045). Note that this time series is relatively similar to the one obtained by scaling the discharge measured at Chiang Saen by the area ratio (equal to 1.17). That time series is depicted by the dotted orange line.

### 4.1.2 Rating curve

With the river cross-section at hand, we estimate the rating curve at the virtual station using the Manning's equation (Equation 1). Since the value of the Manning's coefficient $n$ is unknown, the value of the estimated discharge $Q$ depends not only on the water depth $D$ but also on $n$, that is:

$$Q = \frac{0.161 D^{2.357}}{n}. \tag{6}$$

In Figure 4b, we plot the range of variability of the rating curve corresponding to values of $n$ varying between 0.03 to 0.06 (Section 3.2.2). This range is represented by the light blue band. Note the large increase in river discharge estimates corresponding to a depth larger than 20 m. In this figure, we also report three rating curves corresponding to three specific values of $n$, namely minimum (dotted blue line), average (dark blue line), and maximum (dashed blue line).

### 4.1.3 Remote-sensed discharge

Using the rating curve and water depth (converted from Jason-2/3 altimetry water level data), we estimate 298 discharge data points at the virtual station during the period 2009–2018 (Figure 4c). The light blue band represents the envelope of variability of the discharge corresponding to values of $n$ ranging between 0.03 and 0.06. The figure also depicts the discharge time series corresponding to the average value of the Manning's coefficient ($n = 0.045$), plus an additional time series obtained by scaling the observed discharge at Chiang Saen by a coefficient (equal to 1.17) representing the relative increase in drainage area between Chiang Saen and the virtual station (orange dotted line). A qualitative comparison of these estimated discharge values provides a few useful insights. First, there is large uncertainty in the discharge estimated during the summer monsoon season. This result is explained by the characteristics of the rating curve—the higher the value of $D$, the higher the uncertainty in $Q$ (Figure 4b). Second, there is a larger variability in the discharge estimated during the dry season of 2013-2018 compared to the one of 2009-2012. That is because the cascade dam system in the Lancang modified the natural flow downstream, increasing low flows (Vu et al., 2022). The change can be seen most clearly since 2013, when Nuozhadu, the largest reservoir in the system, became operational. Moreover, it should be considered that the discharge variability could be further amplified by the use of a rating curve—recall that when converting water level to discharge, the higher the water depth value (calculated from water level), the larger the discharge variability (Figure 4b). Lastly, there seems to be a reasonable agreement between the discharge time series corresponding to $n = 0.045$ and the one estimated from values observed at Chiang Sean. This implicitly validates the rating curve, further suggesting that the mean value of $n$ might be a reasonable estimate. To further investigate this last point—and understand how the choice of the Manning's coefficient influences the performance of VIC-Res—we now move to the sensitivity analysis.

## 4.2 Sensitivity analysis

### 4.2.1 Co-dependence between VIC-Res performance and Manning's coefficient

The first fundamental step of our analysis is to understand whether co-estimating the Manning's coefficient and the parameters of the hydrological model (see Figure 1) could bias the calibration process, ultimately limiting the reliability of VIC-Res. To answer this question, we leverage the results obtained by exploring via simulation 1,000 different parameterizations of VIC-Res and Manning's equation.

In Figure 5 (panels (a), (d), (g), and (j)), we illustrate the relationship between the four metrics of performance calculated for VIC-Res (i.e., NSE, TRMSE, MSDE, ROCE) and the value of the Manning's coefficient $n$. To aid the analysis, we highlight in darker color the parameterizations yielding the top 25% performance (250 samples) with respect to each metric. For example, in Figure 5a, the 250 samples with higher NSE are represented by the dark blue lines, while the 750 samples with lower NSE are represented by the light blue lines. The NSE threshold created by the top 25% is equal to 0.48. Interestingly, when comparing these four panels, we see that the values of $n$ corresponding to the best performance vary with the metric we consider. For example, the top 25% performance in terms of NSE is given by values of $n$ ranging between 0.03 and 0.054, while those giving the best performance for MSDE range between 0.037 and 0.06. This point is consolidated by panels (b), (e), (h), and (k), where we show the frequency distribution of $n$ corresponding to the top 25% performance for each metric. The minimum and maximum values we found for each distribution are [0.03, 0.054], [0.034, 0.06], [0.037, 0.06], and [0.033, 0.059] for NSE, TRMSE, MSDE, and ROCE, respectively. Note, also, how the median value of each distribution changes with the selected performance metric.

The explanation behind this result must be sought in the different aspects of the simulated hydrograph that are captured by the four metrics (see Section 3.3.2). Let's consider, for instance, NSE, a metric that emphasizes model performance on high flows: the parameterizations of VIC-Res achieving the top 25% performance are those corresponding to smaller values of $n$, which translate (via the Manning's equation) into higher discharge estimates. In other words, calibrating both models simultaneously while using NSE as a performance metric leads to producing discharge data that are biased towards higher values (panel (c)). Similarly, the values of $n$ associated to the best TRMSE and MSDE performances are shifted upward (i.e. producing lower discharges) as both metrics emphasize model accuracy on lower flows (panels (f) and (i)). For ROCE, most values of $n$ are concentrated around the median value of 0.043 (close to the mean of 0.045). Note that ROCE looks at the overall water balance, thereby requiring calibrating the hydrological models on discharge values that are more centered towards the bulk of the distribution (panel (l)).

### 4.2.2 Breaking the co-dependence

Having established that there can be a co-dependence between the performance of VIC-Res and the Manning's coefficient, we now turn our attention to a potential solution. Ideally, one would like to calibrate a hydrological model that performs well with respect to multiple performance metrics (Efstratiadis and Koutsoyiannis, 2010). Guided by this simple concept, we consider the parameterizations of VIC-Res and Manning's equation associated with the top 25% performance with respect to all metrics

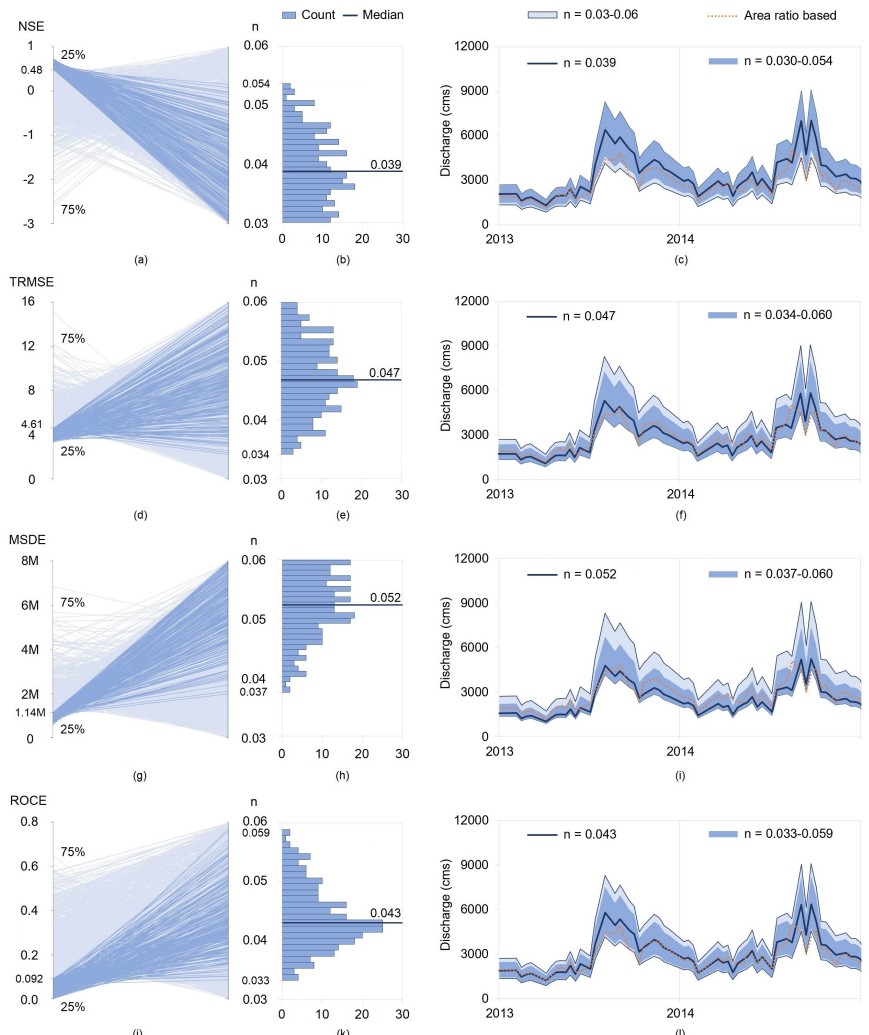

**Figure 5.** The first column contains four parallel-coordinate plots. In each plot, the left axis is a model performance metric (i.e., NSE, TRMSE, MSDE, and ROCE) while the right axis is the Manning's coefficient $n$. Each line corresponds to one of the 1,000 parameterizations generated by Latin Hypercube sampling. The dark blue lines highlight the parameterizations yielding the top 25% performance for each metric. The histograms in the second column illustrate the frequency distribution of $n$ corresponding to these top 25% parameterizations. The median is depicted by the dark blue line. In the last column, we report in light blue the range of variability of the discharge estimated with $n \in [0.03, 0.06]$ (this is the same range as in Figure 4c), and in dark blue the range corresponding to the top 25% performance for each metric. The black lines are the discharge corresponding to the four median values of $n$ (see the second column) and the orange dotted line is the discharge estimated from observations at Chiang Saen via the area-ratio method. Note how the use of different performance metrics results in different ranges and different medians of the Manning's coefficient.

(i.e., NSE, TRMSE, MSDE, and ROCE). This leaves us with 40 parameterizations, illustrated in Figure 6. The first interesting point to note in the figure (right panel) is the empirical distribution of $n$. Focusing on satisfactory performance across multiple metrics yields a narrower range of the Manning's coefficient concentrated around the median value of 0.046. As we shall see later, this means that the uncertainty in discharge values is reduced with respect to what we observed in Figure 5.

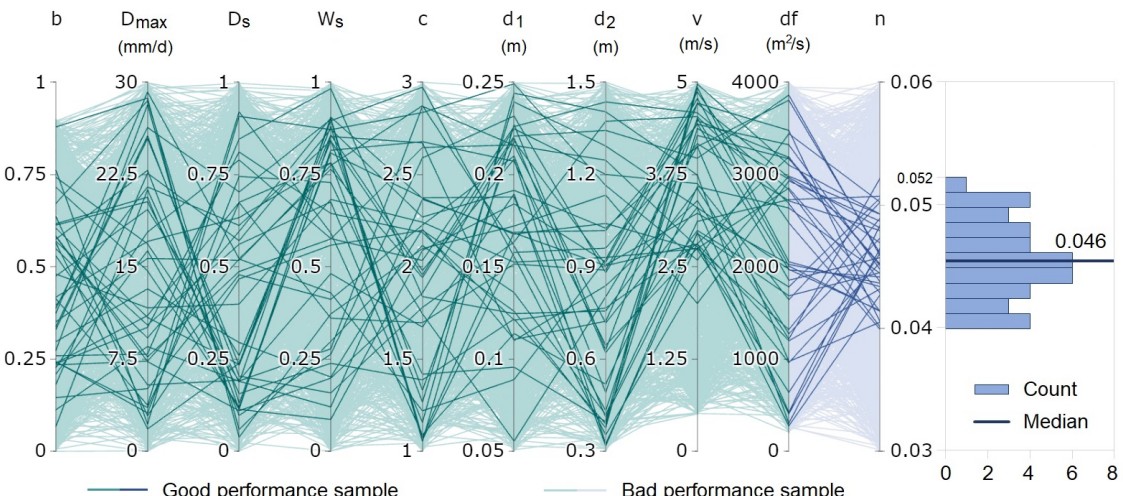

**Figure 6.** Parallel-coordinate plot illustrating the 1,000 parameterizations explored in this study. The first nine axes (green) represent the VIC-Res model parameters, while the last axis (blue) represents the Manning's coefficient $n$. Darker lines highlight the 40 parameterizations showing good performance on all metrics; these are identified by intersecting the four top 25% parameterizations for each performance metric. The panel on the right illustrates the frequency distribution of $n$ corresponding to the 40 selected parameterizations. The median of this distribution is 0.046.

The left panel of Figure 6 illustrates the specific values of the parameterizations through a parallel-coordinate plot, in which each axis represents a parameter and each line is a parameter sample. The 40 top-performing parameterizations are highlighted in bold, while the remaining 960 are depicted with a lighter color. Here, we notice that only the range of the flow velocity ($v$) can be clearly narrowed down (in addition to $n$, of course). Specifically, when considering only the 40 top-performing parameterizations, the range is reduced from $[0.5, 5]$ to $[2, 5]$. For some parameters (i.e., $b$, $d_1$, and $d_2$), although their ranges remain fairly large, the values in the 40 top-performing combinations are mostly (but not exclusively) concentrated in certain parts of the initial range: for $b$, in the middle part (0.2 to 0.75); for $d_1$ in the upper part (0.12 to 0.25), and for $d_2$ in the lower part (0.3 to 1). Lastly, for the remaining parameters (i.e., $D_{max}$, $D_S$, $W_S$, $c$, and $d_f$), the 40 top-performing samples are evenly distributed throughout the initial range. This is a common problem in macro-scale hydrological models, including VIC (Yeste et al., 2020), a point to which we will return in Section 5.

### 4.2.3 Narrowing the uncertainty in discharge data

How does the new parameterization of $n$ impact the remote-sensed discharge data needed to calibrate the model? To answer this question, in Figure 7a we compare two envelops of variability for the discharge data, the one (light blue envelope) corresponding to $n \in [0.03, 0.06]$ (the initial uncertainty range of $n$) and the one (dark blue envelope) corresponding to $n \in [0.04, 0.052]$ (the narrowed range after constraining across all four metrics, see Sec. 4.2.2). As expected, the range of remote-sensed discharge is narrowed down significantly, especially during the high flow periods. Another point that is worth noticing here is that the discharge time series corresponding to the median value of $n$ (i.e., 0.046) is close to the time series estimated from the data available at Chiang Saen. This is a qualitative, yet informative, validation of the sensitivity analysis results.

To complete the analysis, we compare the envelopes of variability of remote-sensed discharge data with the simulations of VIC-Res using the narrow range of $n$ (panel (b)). The comparison shows encouraging results, since the range of simulated discharge (green envelope) is not too wide and it mostly overlap with the remote-sensed one. A good level of overlap is also found in the monthly averages of the simulated and remote-sensed discharge (panel (e)). Looking at specific years (panels (c) and (d)) in more details, reveals more mixed results. In one case (2014), the model predictions seem to follow the estimated discharge very well, particularly in the discharge fluctuations over the summer monsoon. In the other case (2013), instead, the simulated discharge in the wet season (September–November) is $\sim$1.5 to 2.5 times higher than the remote-sensed (and area ratio based) discharge. This may be due to errors in rainfall data used to force the hydrological model, which are common in this region (Kabir et al., 2022).

### 4.3 Model calibration and validation performance

In our last step, we seek to reduce the uncertainty in simulated discharge presented in the previous section. To this purpose, we need to select a specific discharge time series with respect to which we can calibrate the model. Albeit arbitrary, a reasonable choice is the remote-sensed discharge corresponding to the median value of $n$, since (1) it does represent the envelope of variability produced by the Manning's equation and (2) it is rather close to the discharge at the virtual station estimated by scaling the discharge observed at Chiang Saen. Using this time series, we carry out a calibration using the multi-objective evolutionary algorithm described in Section 3.3.3. From the 1,100 Pareto-optimal parameterizations provided by the algorithm, we select the best-performing parameterizations (twelve parameterizations) by applying the same criteria used in the sensitivity analysis (i.e., we take the intersection of the top 25% parameterizations w.r.t. each of the four performance metrics). The envelope of variability of the simulated discharge corresponding to these twelve selected parameterizations is illustrated by the dark green band in Figure 8a, where it is contrasted against the envelope of variability generated by VIC-Res before this calibration step. As expected, the range of variability is narrowed significantly and is well in agreement with the remote-sensed discharge corresponding to a value of $n$ of 0.046 (dark blue line) and the area ratio-based discharge (orange dotted line). The performance metrics of the twelve selected parameterizations—calculated by comparing simulated and remote-sensed discharge at the virtual station—are reasonable, with NSE, TRMSE, MSDE, and ROCE belonging to the ranges [0.686,

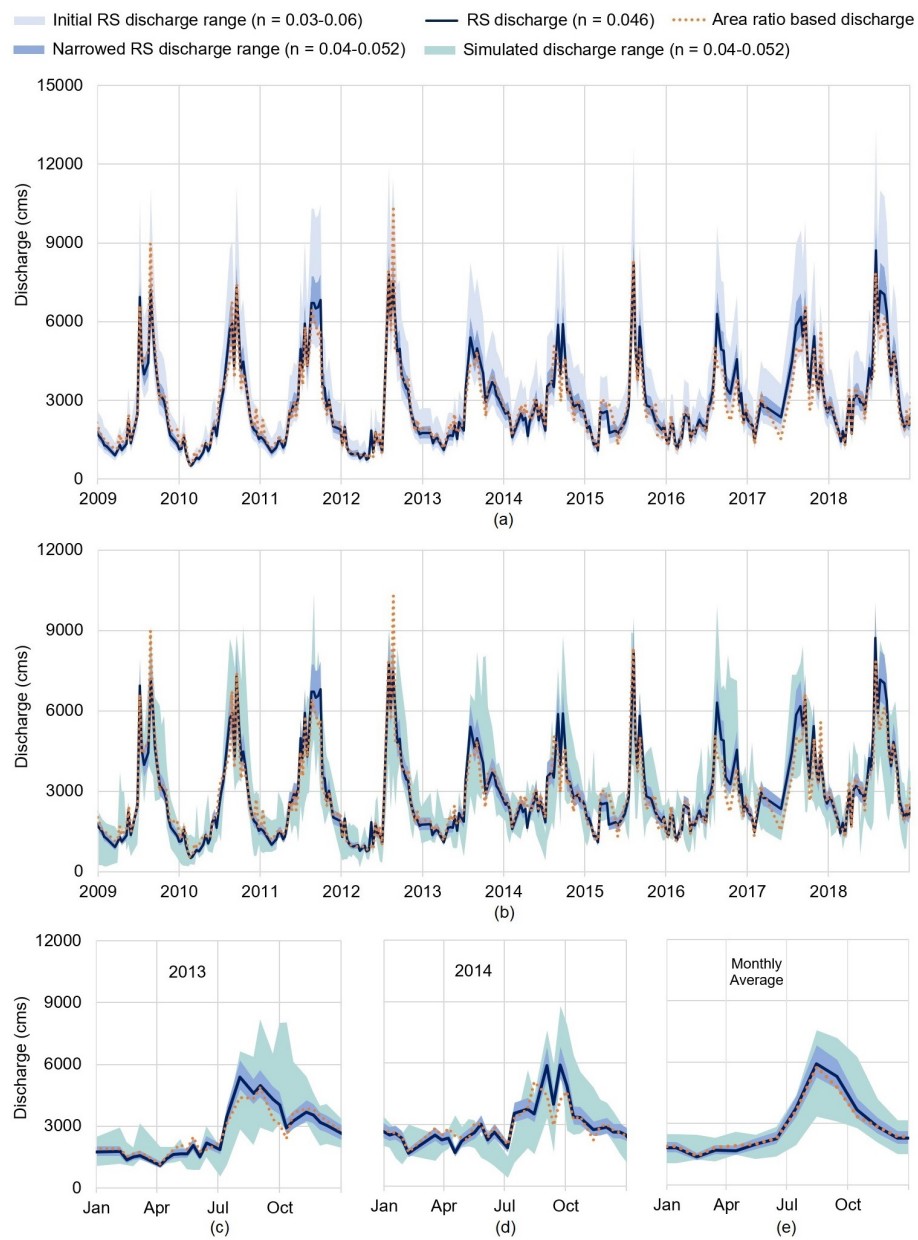

**Figure 7.** In panel (a), we compare the range of variability of the remote-sensed (RS) discharge before and after sensitivity analysis. The two envelopes correspond to values of $n$ belonging to $[0.03, 0.06]$ and $[0.04, 0.052]$. In the plot, we add the discharge values corresponding to the median value of $n$ (0.046) and those estimated from the data at Chiang Saen (orange dotted line). In panel (b), we compare the RS discharge against the discharge data simulated by VIC-Res. Both envelopes correspond to a value of $n \in [0.04, 0.052]$. In panels (c), (d), and (e) we focus on 2013, 2014, and average monthly discharge. The plots of other individual years are provided in Figure S3.

0.689], [3.337, 3.360], [890,904, 908,805], and [0.03, 0.04], respectively. The detailed performance of each parameterization is provided in Table S3.

We report in Figure 8b the performance of the model validation at Chiang Saen station. The variability range of the simulated discharge corresponding to the twelve selected parameterizations (dark green band) is much narrower than the one of the 40 parameterizations selected through the sensitivity analysis (light green band). The new envelope of variability is also well in agreement with the observed discharge at Chiang Saen station (orange dotted line). The performance metrics of the twelve selected parameterizations show only a small decay when compared against those achieved at the virtual station—NSE, TRMSE, MSDE, and ROCE belong to the range [0.594, 0.616], [3.891, 3.935], [1,057,966, 1,071,282], and [0.169, 0.195] respectively. (The detailed performance of each parameterization is provided in Table S4). We note that similar results are achieved by selecting all parameterizations belonging to the Pareto front (58 parameterizations), as shown in Figure S4, Table S5-S6. However, visual inspection of the time series in Fig. 8b shows some discrepancies in the time-to-peak of the discharge at Chiang Saen (e.g., in 2014 and 2017). These discrepancies could be due to different factors including, as already mentioned above, errors in the precipitation data (Kabir et al., 2022). Also note that the comparison of the discharge at Chiang Sean is our validation, we indeed calibrated our model with remote-sensed discharge at the virtual station. Still, overall the fit to observed discharge at Chiang Sean is a remarkable if we consider that the model was calibrated purely with remotely sensed data at the virtual station and no gauged discharge data were used for calibration.

Finally, we looked at the parameter values in the twelve parameterizations selected by the model calibration (Figure S5a). Interestingly, these twelve parameterizations are all quite similar. On the other hand, the 58 parameterizations corresponding to the Pareto front span over a much larger variability range (Figure S5b). Also, these 58 optimal parameterizations are well in agreement with the parameter ranges identified through sensitivity analysis (Section 4.2.2).

## 5 Discussion and Conclusions

Our study contributes an approach for calibrating macro-scale hydrological models in poorly gauged and heavily regulated basins. The approach uses satellite data to infer both the discharge data used for model calibration and the reservoir operations included in the hydrological model. Unlike previous studies, our approach uses Global Sensitivity Analysis to avoid the biases that could be introduced when co-calibrating the hydrological model together with the rating curve used to reconstruct the discharge data (Lima et al., 2019). This fundamental step also helps us narrow down the uncertainty range for the parameterization of the rating curve in a more justified way. In turn, this step paves the way to a more reliable calibration of the hydrological model.

Looking at the specific results of the sensitivity analysis, there are two important points worth stressing here. First, we show that simultaneously estimating the parameters of the hydrological model and the Manning's coefficient (by optimizing a set of model performance metrics) may significantly bias the reconstruction of the discharge values. Different combinations of performance metrics can result in different estimates of river discharge, thereby influencing the parameterization of the hydrological model. We saw, for example, that using NSE for the joint calibration introduces bias in the Manning's coefficient

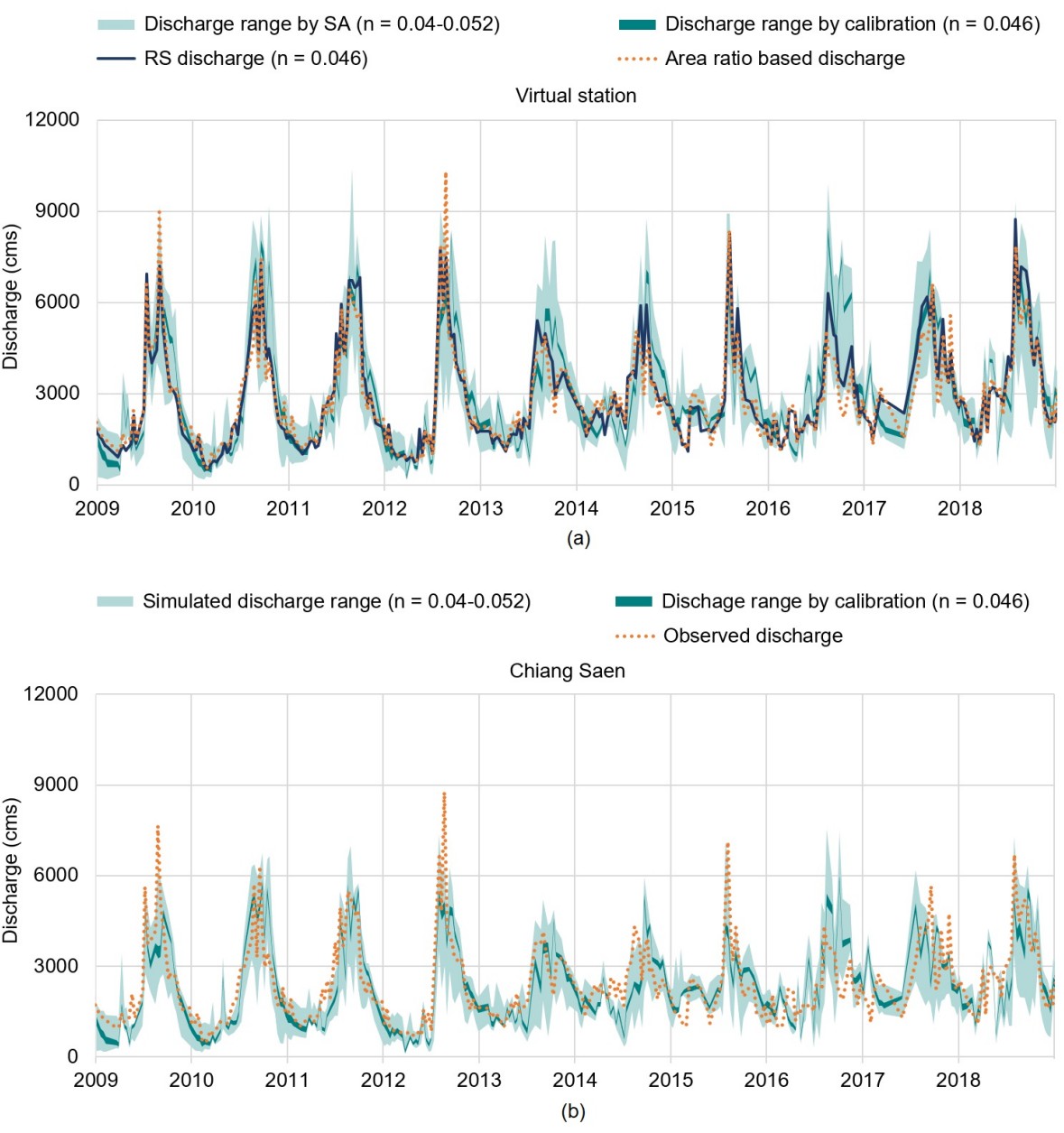

**Figure 8.** Simulated discharge at the virtual station used for calibration (a), and at Chiang Saen station (validation) (b). The dark green band is the variability range of simulated discharge corresponding to the twelve parameterizations produced by the multi-objective calibration, while the light green band corresponds to the 40 parameterizations selected from the sensitivity analysis. The dark blue line is the remote-sensed discharge at the virtual station with $n = 0.046$. In panel (a), the dotted orange line is the discharge at the virtual station scaled from the observed discharge at Chiang Saen. In panel (b), the orange line is the observed discharge at Chiang Saen.

towards producing higher flows. Second, the sensitivity analysis specifically focused on the nine parameters of VIC-Res shows the existence of equifinality, meaning that different parameterizations can yield similar performance in terms of NSE, TRMSE, MSDE, and ROCE. This equifinality issue is perhaps explained by the fact that we are using only river discharge data for calibration. Previous research (Wagener and Pianosi, 2019; Yeste et al., 2020) has shown that a few parameters typically dominate the variability of a given model output (though which parameters that are dominant might vary with the chosen metric). One may therefore expect that observations of other hydrological processes, such as evapotranspiration, could help reduce the uncertainty in the model parameters.

Our numerical framework seeks to reduce the pitfalls hidden in model calibration, but, like any other modelling study, is potentially affected by various errors and uncertainties. First, because of the unavailability of gauged rainfall data, we use a gridded product—a common approach for macro-scale studies. Yet, gridded rainfall data inevitably carry errors, especially in regions, like Southeast Asia, where the number of rainfall gauges is limited (Funk et al., 2015; Kabir et al., 2022). Another potential source of uncertainty is the estimation of the river discharge from altimetry water level data (through a rating curve). Our results show that the estimation is reliable, but in general, the estimation of river cross-section and the use of water surface slope—needed for constructing the rating curve—could contribute to further uncertainty. Specifically, the accuracy of the river cross-section can be affected by the spatial resolution of satellite data (DEM and Landsat images). Also, the DEM, a static 'product' captured at a specific time, hardly grasps the evolution of the river cross-section. Moreover, the interpolation of the cross-section below the lowest observed water level may also cause uncertainty. Meanwhile, there could be uncertainty arising from the estimation of water surface slope from a DEM, which captures the water surface slope at the observation time (i.e., February 2000 for the SRTM). A potential solution to this problem could be to calculate the water surface slope from the time series of altimetry water level at multiple locations nearby the virtual station; an approach that, of course, is only possible when enough data are available. Looking at the issues of river discharge estimation, a potential game changer is the Surface Water and Ocean Topography (SWOT) NASA satellite mission, launched in December 2022. SWOT will provide river width, water level, and water surface slope for major rivers with an average revisit time of 11 days for the next three years (JPL, n.d.). This means we will be able to leverage existing algorithms to estimate river discharge (Gleason and Smith, 2014; Durand et al., 2016; Hagemann et al., 2017) and then inform the implementation of macro-scale hydrological model—an area certainly worth additional research. Yet, we should not forget that model calibration requires time series longer than three years. We could therefore envisage a future in which calibration exercises assimilate multiple discharge data inferred from multiple satellite data.

We note that the approach proposed in this study could be adopted for other basins, although there are a few specific caveats that should be kept in mind. First, the choice of the location for the virtual station (where we construct the river cross-section) should be driven not only by the availability of altimetry data but also by the site topography. In particular, the river banks should not be affected by levees, roads, or other interventions. That is because our approach works best for river banks in natural conditions, where it is possible to infer the relation between river widths and water levels for the portion below the lowest observed water level. Moreover, the virtual station should be located in a straight river segment with minimal discharge variation due to nearby tributaries and distributaries (both upstream and downstream), a setting in which our approach—based

on the Manning's equation—works best (Przedwojski et al., 1995). In such locations, the variation of water surface slope by time is also minimal. Second, if enough data are available, one should consider the option of using time series data of altimetry water level at multiple locations (nearby the virtual station) to estimate water surface slope. Lastly, we note that our modelling approach is applicable to river basins unaffected by the presence of dams; this simply requires to switch-off the reservoir module in VIC-Res (Dang et al., 2020b).

Looking forward, we should consider expanding frameworks like the one presented here to even more complex modelling environments. For example, a modelling challenge that is often recurring in downstream applications is the presence of multiple human interventions, such as dams, irrigation withdrawals, and groundwater pumping. Understanding how data concerning the representation of all these processes influences model calibration remains an open question. A similar comment applies to the calibration of multi-basin and global models. Bringing all these elements together would be a major step towards a more reliable calibration of macro-scale hydrological models.

*Author contributions.* D.T.V., T.D.D., F.P., and S.G. conceptualized the paper and its scope. Data collection and all analyses were carried out by D.T.V. and S.G.. D.T.V wrote the manuscript, with substantial inputs from all authors.

*Code and data availability.* VIC-Res model's codes are available at https://github.com/Critical-Infrastructure-Systems-Lab/VICRes. Reservoir storage data and the python scripts used to produce those data are available at https://github.com/dtvu2205/210520 and https://doi.org/10.5281/zenodo.6299041 (Vu, 2022). Daily discharge data at Chiang Saen were collected from the Mekong River Commission web portal, https://portal.mrcmekong.org/. Altimetry water level data were retrieved from the Database for Hydrological Time Series of Inland Waters (DAHITI), https://dahiti.dgfi.tum.de/. All Landsat images and SRTM-DEM used in our study are available at https://earthexplorer.usgs.gov/.

*Competing interests.* The authors declare that they have no conflict of interest.

*Acknowledgements.* Dung Trung Vu is supported by the SUTD PhD Fellowship.

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
