# Peer review of "Calibrating macro-scale hydrological models in poorly gauged and heavily regulated basins"

_Hydrology and Earth System Sciences, 2023_

## Author Comment (AC1)

**Reply on RC1: 'Comment on hess-2023-35', Anonymous Referee #1**

Overall, the manuscript could be very helpful in providing a guideline for a calibrating model where almost no in-situ data is available. The authors discussed the in-depth methodology of their proposed calibration, providing a detailed analysis of the impact of parameter tuning on the performance metrics of the model. The authors also provided an analysis of the co-dependence of the hydrological and hydraulic model parameterizations and techniques to break the co-dependency. The central argument presented in the manuscript is using satellite data to infer river discharge against which the model-derived river discharge will be compared and the model recalibrated to achieve the desired accuracy. Sattelite data itself can cause a wide range of uncertainty in the calculation of river cross-section, water surface slope, and so on, especially in the Upper Mekong river basin, which is so complex in topography. The satellite data (used as a proxy of observation for calibration purposes) is prone to uncertainty that eventually impact the parameter tuning.  So there's a need to strengthen the discussion by providing a detailed discussion on the uncertainty in the river discharge estimation from satellite data, without which the calibration framework could be questionable.   My recommendation is for a major revision with the specific comments and concerns listed below:

*Response: Thank you for the positive feedback as well as the useful comments for improving the paper.*

1. In the abstract, the authors mentioned that their approach could be readily transferable to another basin. However, the authors did not provide any convincing discussion on how the same framework can be used for another basin. For example, what cautions should other researchers follow when applying the same technique to a highly complex basin with rugged terrain or complex topography? As the estimation of river cross-section and water surface slope could be challenging/more uncertain in some other basin.

*Response: We agree with you. We will add this point to Section 5. Specifically, we will discuss about two key elements concerning the application of our approach (to other basins), namely river cross-section estimation, and water surface slope estimation.*

2. In section 3.3.1 authors discussed the calibrated model parameters and presented the calibration outcomes later. However, one would expect to see a discussion of the calibration of the most sensitive parameters.

*Response: In Section 3.1.1 (Sensitivity analysis), we stated that we carry out a Global Sensitivity Analysis to study the relationship between the performance of VIC-Res and the parameterization of the rating curve (Manning's coefficient). To do that, we investigated a total of 10 model parameters, including 7 soil parameters of the rainfall-runoff module, 2 parameters of the routing module, and the Manning's coefficient. Therefore, in the corresponding result section (4.2, Sensitivity analysis), we focused on providing results related to Manning's coefficient and VIC-Res performance metrics, as they present the co-dependence we are interested in. However, we also showed the results of VIC-Res parameters (Figure 7) and discussed them in the second paragraph of Section 4.2.2. We will expand*

*the discussion on the model parameters in this sub-section. We will also include an analysis of the parameters after the calibration process in Section 4.3 (please refer below for a preliminary version of the analysis).*

[Figure]

*Parameters of the solutions from model calibration. In panel a, solutions were selected by intersecting the four top 25% parameterizations for each performance metric. In panel b, selected solutions are the Pareto solutions.*

3. As the discussion is so central to the simultaneous calibration of the using RS Discharge (from satellite data). Thus an uncertainty analysis of the estimation of the river cross-section or uncertainty in the RS Discharge from satellite data is extremely necessary. It directly impacts the RS Discharge estimation against which VIC-RES discharge is compared to calculate performance matrices. Thus uncertainty in RS Discharge can substantially impact the calibration process and parameter tuning, fundamentally questioning the Novel technique the authors suggested in that manuscript. I recommend a discussion on uncertainty in RS Discharge estimation and how it may affect the calibration process. Although the author provided some insights in section 4.2.3

*Response: Besides some insights provided in sections 4.1.3 (we believe that you meant Section 4.1.3 instead of 4.2.3), we discussed the uncertainty in RS discharge in the third paragraph of Section 5. This said, we agree with you that this point is important, so we will proceed by expanding and strengthening it.*

4. In section 3.3.2- The authors discussed that they used multiple performance matrices to cover a different aspect of modeling accuracy. However, the use of KGE as a performance metric is also suggested, as it considers bias, correlation, and variability.

*Response: Thanks for your suggestion. Here, we provide the result using KGE. Noticeably, using KGE yields results similar to those obtained with NSE. In particular, the top 25% parameterizations w.r.t. KGE (250 samples) have 208 parameterizations in common with those of NSE, thereby resulting in similar narrowed ranges of Manning's coefficient (n) and RS discharge. The reason for this result is probably due to the fact that KGE is based on a decomposition of NSE into correlation, variability bias, and mean bias components (Gupta et al., 2009). Because of the similarity in results obtained with NSE and KGE, we decide to exclude the ones associated to KGE in our analysis.*

[Figure]

*In panel a, the dark blue lines highlight the parameterizations yielding the top 25% performance (highest KGE). The histogram in panel b illustrates the frequency distribution of n corresponding to the top 25% parameterizations with the median depicted by the dark blue line. In panel c, the light blue envelop is the range of variability of the discharge estimated with n $\in$ [0.03,0.06], while the dark blue envelop is the range corresponding to the top 25% performance. The black lines are the discharge corresponding to the median values of n, while the orange dotted line is the discharge estimated from observations at Chiang Saen via the area-ratio method.*

5. In Figure 8: could you discuss why there is less variability in 2009-2012 and after that, there is considerable variability, particularly in the low flows?

*Response: This is a very good point. First, note that the cascade dam system in the Upper Mekong modified the natural flow downstream, it increased low flows (Vu et al., 2022). The change can be seen most clearly since 2013, when Nuozhadu—the largest reservoir in the Upper Mekong—became operational. This change is captured in the altimetry water level, which we use to convert to RS discharge through the rating curve. On the other hand, as shown in Figure 5b, when converting from water depth (calculated from water level) to discharge, the higher the value of water depth, the wider the discharge variability. That explains the considerable variability in RS discharge of 2013-2018 compared to the one of 2009-2012. We will add this point to Section 4.1.3.*

6. In Figure 9: the timing of the peak is missed in some of the years, e.g., 2007. You can just add a discussion on the sensitivity of different parameter tuning in capturing timing/seasonality or peak. Or which is the most sensitive parameter?

*Response: Overall, the RS, simulated, and area-ratio discharges at the virtual station (Figure 9a) show similar behaviors in the time-to-peak, so we believe you refer to the comparison against the observed discharge at Chiang Saen (Figure 9b), where some discrepancies in the time-to-peak emerge (e.g., in 2014 and 2017). These discrepancies could be due to different factors. First, and most important, note that we calibrated our model with RS discharge at the virtual station and then validated it with observed discharge at Chiang Sean. Another reason could be the uncertainty due to the use of gridded precipitation data (Kabir et al., 2022). We will elaborate on these points in Section 4.3.*

7. Also. How can the hydrological response unit's resolution or size impact calibration? It can impact the calibration substantially. For example, the Lancang river basin is so narrow and elongated. Thus, the use of a coarse hydrological response unit of Coarse-resolution may accurately impact the identification of river grid cells.

*Response: The hydrological response unit's resolution (or size of distributed/grided hydrological models) could affect the rainfall-runoff and routing estimations, and thus affect simulated discharge and model calibration (Egüen et al., 2012). Looking at the existing distributed models for the Mekong region, we note that our cell size (i.e., 0.0625°) somewhat falls in between w.r.t. what is currently being adopted—for example, Costa-Cabral et al. (2007) and Tatsumi & Yamashiki (2015) adopted a resolution of 1/12° and 0.25°, while Du et al. (2020) and Bonnema and Hossain (2017) used a resolution of 90m/900m and 0.01°, respectively. We agree that this point is important, but also believe that it applies to 'any' modelling exercise, not only to those relying on remotely-sensed data, like ours. Because of this reason, we feel that adding a thorough analysis on the impact of model resolution would go beyond the goal of our study—and potentially confuse its main message. We thus suggest including a discussion on this point in Section 5.*

8. Figure 2 and 3 can be merged together. Having two figures does not add much value to the discussion.

*Response: We agree with you on this point. We will condense Figure 2 and 3 into a new figure.*

9. In section 3.2.1: The authors used a regression technique (sixth-degree polynomial) to fit the data point best. However, the author said that is best works for the natural condition of rivers. AS MANUSCRIPT TITLE, the authors mentioned "Heavily Regulated Basin." One would like to know the author's novel technique for heavily regulated basins. In the suggested numerical framework, I think I do not see any strong linkage of the reservoir operation (like heavy regulation) with the calibration/parameter tuning. Or could you provide an explanation of how your technique is mainly applicable to the heavily regulated basin? Or it may be more justifiable to say Novel calibration technique for the poorly gauged basin.

*Response: Thank you for raising this point. Let us begin by clarifying a point that is perhaps at the origin of this comment: As stated in the title of the manuscript, our framework is developed to calibrate models in heavily regulated basins (that is, where river discharge is modified from its natural flow by man-made reservoirs). Part of this framework includes a method to construct the river cross-section at a virtual station (Section 3.2.1). We stated that this specific method works best for riverbanks (topography) in natural conditions. We do not see any conflict between our overall intent and the specific method used to infer discharge, because the term "heavily regulated" is used to describe flow regime while the term "natural conditions" is used to refer to the river topography at a specific location.*

*Moving to the choice of the manuscript title (as well as our contribution), we would like to begin by noticing that reservoir operations could affect the model parameterization. Calibrating hydrological models with and without the representation of reservoirs could result in different sets of model parameters, even though they both have good model performances (i.e., comparable simulated and observed discharges). This is because, when reservoir operations are not included, model calibration adopts parameterizations that compensate for the absence of the reservoirs (Dang et al., 2020). In our case study, the discharge (observed or inferred from satellite data) includes the flow modification caused by reservoirs, so the reservoir operations must be included in the model. Specifically, we do that by integrating the reservoir operations estimated from satellite data (to replace the measured data due to their unavailability) into the hydrological model. To the best of our knowledge, this specific approach is still at its infancy in the hydrological modelling domain, so this is why we would like to retain the term "heavily regulated basins" in the title.*

*Finally, we note (and agree with the reviewer) that our approach is not only applicable to regulated basins. In fact, the reservoir operation module could be switched off when working on a natural catchment. We will elaborate on this point in Section 5.*

*References*

Bonnema, M. and Hossain, F.: Inferring reservoir operating patterns across the Mekong Basin using only space observations, Water Resources Research, 53, 3791-3810, https://doi.org/10.1002/2016wr019978, 2017.

Costa-Cabral, M. C., Richey, J. E., Goteti, G., Lettenmaier, D. P., Feldkötter, C., and Snidvongs, A.: Landscape structure and use, climate, and water movement in the Mekong River basin, Hydrological Process, 22, 1731-1746, https://doi.org/10.1002/hyp.6740, 2007.

Dang, T. D., Chowdhury, A. F. M. K., and Galelli, S.: On the representation of water reservoir storage and operations in large-scale hydrological models: Implications on model parameterization and climate change impact assessments, Hydrology and Earth System Sciences, 24, 397–416, https://doi.org/10.5194/hess-24-397-2020, 2020.

Du, T. L. D., Lee, H., Dui, D. D., Arheimer, B., Li, H., Olsson, J., Darby, S. E., Sheffield, J., Kim, D., and Hwang, E.: Streamflow prediction in "geopolitically ungauged" basins using satellite observations and regionalization at subcontinental scale, Journal of Hydrology, 588, 125016, https://doi.org/10.1016/j.jhydrol.2020.125016, 2020.

Egüen, M., Aguilar, C., Herrero, J., Millares, A., and Polo, M. J.: On the influence of cell size in physically-based distributed hydrological modelling to assess extreme values in water resource planning, Natural Hazards Earth System Sciences, 12, 1573-1582, https://doi.org/10.5194/nhess-12-1573-2012, 2012.

Gupta, H. V., Kling, H., Yilmaz, K. K., and Martinez, G. F.: Decomposition of the mean squared error and NSE performance criteria: Implications for improving hydrological modelling, Journal of Hydrology, 377, 80-91, https://doi.org/10.1016/j.jhydrol.2009.08.003 2009.

Kabir, T., Pokhrel, Y., and Felfelani, F.: On the precipitation-induced uncertainties in process-based hydrological modeling in the Mekong River Basin, Water Resources Research, 58, e2021wr030 828, https://doi.org/10.1029/2021wr030828, 2022.

Tatsumi K. and Yosuke Yamashiki, Y.: Effect of irrigation water withdrawals on water and energy balance in the Mekong River Basin using an improved VIC land surface model with fewer calibration parameters, Agricultural Water Management, 159, 92-106, http://dx.doi.org/10.1016/j.agwat.2015.05.011, 2015.

Vu, D. T., Dang, T. D., Galelli, S., and Hossain, F.: Satellite observations reveal 13 years of reservoir filling strategies, operating rules, and hydrological alterations in the Upper Mekong River basin, Hydrology and Earth System Sciences, 26, 2345-2364, https://doi.org/10.5194/hess-26-2345-2022, 2022.

---

## Author Comment (AC2)

**Reply on RC2: 'Comment on hess-2023-35', Andrea Galletti,**

The authors propose an interesting framework for exploiting satellite data for the calibration of large-scale hydrological model. The steps undertaken to translate satellite-derived information into are explained in thorough detail. Then, the co-dependence between the hydrological model's performance and the calibration of the hydraulic model based on remote-sensed data is analyzed via Global Sensitivity Analysis. Finally, the authors adopt the satellite-derived information to calibrate VIC-Res and validate the predictions against available ground observation.

The concept is promising, however it would benefit from some considerations on its general applicability were drawn. Furthermore, the densely-flowing work sometimes overshadows the connection between the overall proposed framework and the particular tool/result being explained. I recommend acceptance of the paper, provided that the authors address or clarify the following points (minor revision):

*Response: Thank you for the positive feedback as well as the useful comments for improving the paper.*

1. The paper addresses the issue of calibrating macro-scale models in ungauged catchments. The introduction seems to implicitly assume that every large-scale hydrological model is always calibrated, and that this is most times done against available streamflow time series. However, several large scale hydrological models do not actually undergo case-specific calibration. While a calibrated model can provide results that are generally more reliable or realistic, this might not be clear to every reader, or someone could disagree. I recommend addressing the importance and effectiveness of calibration at the beginning (lines up to 15).

*Response: We agree with you. We will make this point clear at the beginning of Introduction.*

2. The subsequent paragraph (line 17 onwards) could also benefit from a more thorough introduction to models' calibration and what is needed. In particular, it is not clearly explained why and how the presence of hydropower infrastructures for which operations are not known can create pitfalls for model calibration, nor why could it be at all important to include such factors in a calibration of hydrological model's parameters.

*Response: We agree with you. We will address these points in the Introduction. Specifically, we will explain how the presence/absence of hydropower infrastructures (reservoir operations) affects model parameterization and the importance of including such factor in the model calibration process.*

3. Some references and examples should be provided to support the sentence "Yet, this approach may still partially rely on in-situ data": this sentence opens up to the second class of RS-based works and basically justifies the study, therefore it needs to be backed by clear examples.

*Response: We provided some references/examples in the sentence right before (line 34). We realized that our expression could cause a misunderstanding. We will revise it accordingly.*

4. Following point #1, I feel like a non-trivial question is whether it is possible and helpful to calibrate large scale hydrological models in ungauged catchments. This opens up to research questions I, II and III, helping to place them in a context that is broader than a mere numerical exercise.

*Response: Thank you for this comment, which pushed us to think more about the broader context of our work. As already pointed out in point #1, several large-scale hydrological models do not actually undergo case-specific calibration. We know that this can often lead to poor model performances. Hence, calibration has been suggested as a key avenue for model improvement---whenever computationally feasible (e. g., Bierkens et al., 2015; Samaniego et al., 2017). However, a prerequisite for a "successful" calibration is the availability of "good quality" data (i.e., at a minimum, "sufficiently long" time series of discharge observations and with "sufficiently small" observational errors). In many regions of the world, such good quality data are not available from in-situ observation networks, so the question arises whether remote-sensed data, despite all their uncertainty, are still "good enough" to support model calibration. Hence, the overarching question that this paper addresses is: to what extent is it possible and helpful to calibrate a large-scale hydrological model in ungauged catchments using remote-sensed data? How do we deal with potential interactions between parameters used in data pre-processing (i.e., from remote-sensed data to reconstructed discharge data) and parameters of the hydrological models when doing model calibration? Can we reduce uncertainty in model calibration results by properly taking into account such interactions? We will make sure that the overarching question underpinning our work is clearly illustrated in the Introduction.*

5. The choice of the 2009-2018 time window is not motivated as of line 84. I suspect the decision was taken backwards for compatibility with data coverage and quality (Figure 3). If this is the case, it should be stated clearly and the corresponding statements (line 96 and caption of fig 3) adjusted. If the choice was driven by something else, it should be pointed out.

*Response: Yes, the reason is not only to include the filling period of the two largest reservoirs (as stated in line 83-84) but also to account for coverage and temporal resolution of altimetry data. We will make these points clearer.*

6. I am dubious about the reliability of interpolating (linearly) daily reservoir operations between two monthly values. Hydropower often operates at the daily scale (or lower). The configuration of cascading dams might help masking errors in this procedure and their repercussions on the performance metrics. Did the authors compare their results with a fully-natural setup in order to evaluate the impacts of the reconstructed reservoir operation?

*Response: To answer this comment, let us first elaborate on how reservoirs in the Upper Mekong River have been designed and operated. Xiaowan and Nuozhadu are the two largest reservoirs: they have a massive capacity (~36 km³) and account for about 85 % of the total system's storage. Because of their size, their role is not to follow inter and intra-daily electricity demand variability, but rather to ensure a stable supply of power and to minimize the variability in the production of the other dams composing the hydropower system. This goal is reflected by their operating patterns. In the wet season (June-November), Xiaowan and Nuozhadu reservoirs gradually store water until reaching their maximum operational level (and release extra water if necessary). The other reservoirs run at their normal operational level (full capacity for power generation). In the dry season (December-May), Xiaowan and Nuozhadu gradually release water to the downstream reservoirs to ensure that the other reservoirs can run at their normal operational level. Putting this information together, we believe it is fair to state that Xiaowan and Nuozhadu are characterized by slow-varying dynamics (International Rivers, 2014; Vu et al., 2022). Hence, basing our analysis on interpolated monthly values is reasonable, although not ideal, of course. Our argument is partially supported by the analysis carried out in Vu et al. (2022), where we successfully compared the monthly storage of Xiaowan and Nuozhadu derived from Landsat images against the storage derived from Jason altimetry data (10-day temporal resolution) and Sentinel-1/2 images (6-day temporal resolution). Because of the spatial and temporal coverages of those data, we use the result derived from Landsat images for this study. We understand that all the information mentioned above is important. We will add it to Section 3.1.2.*

*A comparison between the river discharge simulated with and without reservoirs is technically feasible (see the figure below). However, we do not fully understand how this comparison can help evaluate the process of interpolating daily reservoir operations between two monthly values, and thus we believe that the explanation provided above is more compelling. Hence, we suggest including this in the response to reviewers but not in the revised manuscript.*

[Figure]

*Comparisons between simulated discharges with dams (dark green) and without dams (yellow green) at the virtual station (a) and Chiang Saen station (b). The variation ranges of both two settings (with and without dams) are corresponding to the twelve selected solutions in calibration (explained in Section 4.3).*

7. The estimation of the river cross section represents an important step in the framework, influencing the outcomes of the rating curve and all subsequent analyses. With the river width ranging from 400m to 200m depending on water level, and with DEM and Landsat images having a resolution of 30m, one could think that the combination of cells taken as the cross section is not univoque. Exploring multiple alternatives for the chosen cross section could help assessing the variability of this extrapolation (if at all present). Otherwise, a plane view or schematic of the chosen pixels would help understanding the decision taken. Furthermore, river bed might be subjected to vertical evolution (i.e., erosion or sediment accumulation) which is hardly grasped by this methodology since it relies on a static DEM. Performing a similar extrapolation (of the river cross section) e.g.,at location 812 which is closer to Chiang Saen and then comparing the so-obtained discharge values with ground observations would have provided a rather solid, yet not exhaustive, base for any assumption made at Virtual Station, reinforcing the implicit validation described at lines 267-269.

*Response: Thank you for the suggestion of exploring multiple alternatives for the chosen cross-section. In the panels a-d of the figure below, we show the results of four alternative cross-sections, created by moving the one at the location of the virtual station (reported in the manuscript, marked by VS in the panel e and f) 30 and 60m (1 and 2 cells) both upstream and downstream. The alternatives are well in agreement with the one reported in the manuscript. Specifically, riverbed elevations are 277.2, 275.6, 276, 274.5, and 274.3 m a.s.l. (from upstream to downstream). We plot the five cross-sections together in panel e, and show a 3D visualization in panel f. We plan to add this part to the Supplement.*

*We agree with the second point that riverbeds might be subjected to vertical evolution, which is hardly grasped by a static DEM. We will discuss this point as one of the limitations of the methodology (in Section 5).*

*As for the last point, we believe that applying our methodology to location 812 and other locations which are close to Chiang Saen station may not be possible. This is due to a number of issues: (1) the temporal resolution (35 and 27 days for Envisat and Sentinel-3A respectively) and coverage (see Figure 3) of altimetry data is too coarse; (2) Manning's equation works best for straight river segments with limited discharge variations due to tributaries and distributaries nearby (Przedwojski et al., 1995), while most locations close to Chiang Saen are located in curved river segments, where many tributaries join the mainstream of the Mekong River; and (3) modifications of the riverbank topography (for road and agriculture/farming) around most of those locations affect river cross-section estimations. We also plan to elaborate on these points in Section 2.3 and Section 5, where we will expand the discussion on the limitations of our methodology.*

[Figure]

*Panels a-d show the results of four alternative cross-sections, created by moving the one at the location of the virtual station (reported in the manuscript, marked by VS in the panel e and f) 30 and 60m (1 and 2 cells) both upstream and downstream. The five cross-sections are plotted together in panel e, and a 3D visualization is provided in panel f.*

8. Line 338 onwards: the selected 25% is said to be the best according to all metrics: does this mean that the (i assume normalized, as defined in the methods) metrics were averaged and the best 25% averages were taken? This concept could be made a bit clearer. Furthermore, it is unclear what determined the choice of the 12 and 58 stations presented later in this section.

*Response: To select the solutions we first created four groups (each containing the top 25% solutions w.r.t. a single performance metric) and then took their intersection. This approach yields 12 solutions. The other approach we considered is the definition of Pareto efficiency, which yields 58 solutions. We will make sure this point is crystal clear to avoid any misunderstandings.*

9. Line 359: Phrasing here is a bit misleading, and a more thorough introduction could be useful: GSA was used to determine the co-dependence between n and the model's performance, while the potential bias intrinsic to the co-dependence was not really assessed, although one could reasonably suppose that it is present. Providing further evidence (references) that joint estimation of discharge and parameters leads to a biased calibration could help strenghten the need to break the co-dependence.

*Response: Yes, we agree with you that this part is potentially misleading, so we will revise it accordingly. We note that the existing literature does not look specifically at the problem of biased calibration (when co-calibrating a hydraulic and hydrological model). In fact, this is, to the best of our knowledge, the first study that looks into the pitfalls of such co-calibration process.*

TECHNICAL CORRECTIONS:

Line 227: is 2008-2018 correct? it seems it should be 2009
*Response: Yes. You are right. It is 2009-2018.*

Line 297: the value in parentheses should be 0.045
*Response: Yes. You are right. It is 0.045.*

Line 382-383: do not contribute to (reducing) modelling uncertainty?
*Response: It is "could contribute to modelling uncertainty".*

Line 393: One often recurring... one should be removed?
*Response: Yes, we will improve the readability of this sentence.*

*References*

Bierkens, M. F. P.: Global hydrology 2015: State, trends, and directions, Water Resources Research, 51, 4923-4947, https://doi.org/10.1002/2015wr017173, 2015.

International Rivers: The Environmental and Social Impacts of Lancang Dams, https://archive.internationalrivers.org/sites/default/files/attached-files/ir_lancang_dams_researchbrief_final.pdf, last access 11 April 2023, 2014.

Przedwojski, B., Blazejewski, R., Pilarczyk, K.: River training techniques: Fundamentals, design and applications, Netherlands: Taylor & Francis, 1995.

Samaniego, L., Kumar, R., Thober, S., Rakovec, O., Zink, M., Wanders, N., Eisner, S., Schmied, H. M., Sutanudjaja, E. H., Warrach-Sagi, K., and Attinger, S.: Toward seamless hydrologic predictions across spatial scales, Hydrology Earth System Sciences, 21, 4323-4346, https://doi.org/10.5194/hess-21-4323-2017, 2017.

Vu, D. T., Dang, T. D., Galelli, S., and Hossain, F.: Satellite observations reveal 13 years of reservoir filling strategies, operating rules, and hydrological alterations in the Upper Mekong River basin, Hydrology and Earth System Sciences, 26, 2345-2364, https://doi.org/10.5194/hess-26-2345-2022, 2022.

---

## Author Response (AR1)

**Reply on RC1: 'Comment on hess-2023-35', Anonymous Referee #1**

Overall, the manuscript could be very helpful in providing a guideline for a calibrating model where almost no in-situ data is available. The authors discussed the in-depth methodology of their proposed calibration, providing a detailed analysis of the impact of parameter tuning on the performance metrics of the model. The authors also provided an analysis of the co-dependence of the hydrological and hydraulic model parameterizations and techniques to break the co-dependency. The central argument presented in the manuscript is using satellite data to infer river discharge against which the model-derived river discharge will be compared and the model recalibrated to achieve the desired accuracy. Sattelite data itself can cause a wide range of uncertainty in the calculation of river cross-section, water surface slope, and so on, especially in the Upper Mekong river basin, which is so complex in topography. The satellite data (used as a proxy of observation for calibration purposes) is prone to uncertainty that eventually impact the parameter tuning. So there's a need to strengthen the discussion by providing a detailed discussion on the uncertainty in the river discharge estimation from satellite data, without which the calibration framework could be questionable. My recommendation is for a major revision with the specific comments and concerns listed below:

*Response: Thank you for the positive feedback as well as the useful comments for improving the paper. We addressed all comments listed below; in particular, we improved the presentation of our results, clarified a few technical points, and, very important, expanded Section 5 to discuss about the limits and uncertainties associated to the proposed approach.*

1. In the abstract, the authors mentioned that their approach could be readily transferable to another basin. However, the authors did not provide any convincing discussion on how the same framework can be used for another basin. For example, what cautions should other researchers follow when applying the same technique to a highly complex basin with rugged terrain or complex topography? As the estimation of river cross-section and water surface slope could be challenging/more uncertain in some other basin.

*Response: We agree with you. We added a new paragraph (the fifth paragraph in Section 5, Lines 484-495 in the mark-up version of the revised manuscript) where we discussed about the key elements concerning the application of our approach to other basins, including river cross-section estimation and water surface slope estimation. We also briefly touched upon the application of our approach to natural river basins.*

2. In section 3.3.1 authors discussed the calibrated model parameters and presented the calibration outcomes later. However, one would expect to see a discussion of the calibration of the most sensitive parameters.

*Response: In Section 3.1.1 (Sensitivity analysis), we stated that we carry out a Global Sensitivity Analysis to study the relationship between the performance of VIC-Res and the parameterization of the rating curve (Manning's coefficient). To do that, we investigated a total of 10 model parameters, including 7 soil parameters of the rainfall-runoff module, 2 parameters of the routing module, and*

*the Manning's coefficient. Therefore, in the corresponding result section (4.2, Sensitivity analysis), we focused on providing results related to Manning's coefficient and VIC-Res performance metrics, as they present the co-dependence we are interested in. However, we also showed the results of VIC-Res parameters in Figure 7 (Figure 6 in the revised manuscript) and discussed them in the second paragraph of Section 4.2.2. We have now expanded the discussion of the model parameters in this sub-section (Lines 579-583 in the mark-up version of the revised manuscript). Moreover, we included a discussion of the parameters after the calibration process in Section 4.3 (Lines 435-438 in the mark-up version of the revised manuscript) and added the plot below to the Supplement (Figure S5).*

[Figure]

*Figure S5. VIC-Res parameters after model calibration. In panel (a), solutions were selected by intersecting the four top 25% parameterizations for each performance metric. In panel (b), the highlighted parameterizations correspond to the Pareto solutions.*

3. As the discussion is so central to the simultaneous calibration of the using RS Discharge (from satellite data). Thus an uncertainty analysis of the estimation of the river cross-section or uncertainty in the RS Discharge from satellite data is extremely necessary. It directly impacts the RS Discharge estimation against which VIC-RES discharge is compared to calculate performance matrices. Thus uncertainty in RS Discharge can substantially impact the calibration process and parameter tuning, fundamentally questioning the Novel technique the authors suggested in that manuscript. I recommend a discussion on uncertainty in RS Discharge estimation and how it may affect the calibration process. Although the author provided some insights in section 4.2.3

*Response: We addressed this comment in two complementary ways. First, we added a few more insights concerning the estimation of remote-sensed discharge in Section 4.1.3 (Lines 328-333 in the mark-up version of the revised manuscript). Second, we expanded and strengthened the discussion in Section 5 (Lines 462-474 in the mark-up version of the revised manuscript).*

4. In section 3.3.2- The authors discussed that they used multiple performance matrices to cover a different aspect of modeling accuracy. However, the use of KGE as a performance metric is also suggested, as it considers bias, correlation, and variability.

*Response: Thanks for your suggestion. Here, we provide the results using KGE. Noticeably, using KGE yields results similar to those obtained with NSE. In particular, the top 25% parameterizations w.r.t. KGE (250 samples) have 208 parameterizations in common with those of NSE, thereby resulting in similar narrowed ranges of Manning's coefficient (n) and RS discharge. The reason for this result is probably due to the fact that KGE is based on a decomposition of NSE into correlation, variability bias, and mean bias components (Gupta et al., 2009). Because of the similarity in results obtained with NSE and KGE, we decide to exclude the ones associated to KGE in our analysis.*

[Figure]

*In panel (a), the dark blue lines highlight the parameterizations yielding the top 25% performance (highest KGE). The histogram in panel (b) illustrates the frequency distribution of n corresponding to the top 25% parameterizations with the median depicted by the dark blue line. In panel (c), the light blue envelop is the range of variability of the discharge estimated with n $\in$ [0.03,0.06], while the dark blue envelop is the range corresponding to the top 25% performance. The black lines are the discharge corresponding to the median values of n, while the orange dotted line is the discharge estimated from observations at Chiang Saen via the area-ratio method.*

5. In Figure 8: could you discuss why there is less variability in 2009-2012 and after that, there is considerable variability, particularly in the low flows?

*Response: This is a very good comment. First, note that the cascade dam system in the Upper Mekong modified the natural flow downstream; in particular, it increased low flows (Vu et al., 2022). The change can be seen most clearly since 2013, when Nuozhadu—the largest reservoir in the Upper Mekong—became operational. Second, this change is captured by the altimetry water level, which is converted into RS discharge through the rating curve. This is important because, as shown in Figure 5b (Figure 4b in the revised manuscript), when converting from water depth (calculated from water level) to discharge, the higher the value of water depth, the larger the discharge variability. These two points together explain the considerable variability in RS discharge of 2013-2018 compared to the one of 2009-2012. We added this comment to Section 4.1.3 (Lines 328-333 in the mark-up version of the revised manuscript).*

6. In Figure 9: the timing of the peak is missed in some of the years, e.g., 2007. You can just add a discussion on the sensitivity of different parameter tuning in capturing timing/seasonality or peak. Or which is the most sensitive parameter?

*Response: Overall, the RS, simulated, and area-ratio discharges at the virtual station in Figure 9a (Figure 8a in the revised manuscript) show similar behaviors in the time-to-peak, so we believe you refer to the comparison against the observed discharge at Chiang Saen in Figure 9b (Figure 8b in the revised manuscript), where some discrepancies in the time-to-peak emerge (e.g., in 2014 and 2017). These discrepancies could be due to different factors. First, and most important, note that we calibrated our model with RS discharge at the virtual station and then validated it with observed discharge at Chiang Sean. Another reason could be the uncertainty due to the use of gridded precipitation data (Kabir et al., 2022). We elaborated on these points in Section 4.3 (Lines 428-431 in the mark-up version of the revised manuscript).*

7. Also. How can the hydrological response unit's resolution or size impact calibration? It can impact the calibration substantially. For example, the Lancang river basin is so narrow and elongated. Thus, the use of a coarse hydrological response unit of Coarse-resolution may accurately impact the identification of river grid cells.

*Response: Yes, the resolution of the hydrological response unit's (or size of distributed/grided hydrological models) could affect the rainfall-runoff and routing estimations, and thus affect simulated discharge and model calibration (Egüen et al., 2012). Looking at the existing distributed models for the Mekong region, we note that our cell size (i.e., 0.0625°) somewhat falls in between what is currently being adopted—for example, Costa-Cabral et al. (2007) and Tatsumi & Yamashiki (2015) adopted a resolution of 1/12° and 0.25°, while Du et al. (2020) and Bonnema and Hossain (2017) used a resolution of 90 m/900 m and 0.01°, respectively. We agree that this point is important, but also believe that it applies to 'any' modelling exercise, not only to those relying on remote-sensed data, like ours. Because of this reason, we feel that adding a thorough analysis on the impact of model resolution would go beyond the goal of our study—and potentially confuse its main message. We thereby preferred to discuss the choice of cell size in Section 3.1.1 (Lines 172-178 in the mark-up version of the revised manuscript).*

8. Figure 2 and 3 can be merged together. Having two figures does not add much value to the discussion.

*Response: We agree with you on this point. We condensed Figures 2 and 3 into a new figure (Figure 2 in the revised manuscript).*

9. In section 3.2.1: The authors used a regression technique (sixth-degree polynomial) to fit the data point best. However, the author said that is best works for the natural condition of rivers. AS MANUSCRIPT TITLE, the authors mentioned "Heavily Regulated Basin." One would like to know the author's novel technique for heavily regulated basins. In the suggested numerical framework, I think I do not see any strong linkage of the reservoir operation (like heavy regulation) with the calibration/parameter tuning. Or could you provide an explanation of how your technique is mainly applicable to the heavily regulated basin? Or it may be more justifiable to say Novel calibration technique for the poorly gauged basin.

*Response: Thank you for raising this point. Let us begin by clarifying a point that is perhaps at the origin of this comment: As stated in the title of the manuscript, our framework is developed to calibrate models in heavily regulated basins (that is, where river discharge is modified from its natural flow by man-made reservoirs). Part of this framework includes a method to construct the river cross-section at a virtual station (Section 3.2.1). We also stated that this specific method works best for riverbanks (topography) in natural conditions. This said, we do not see any conflict between our overall intent and the specific method used to infer discharge, because the term "heavily regulated" is used here to describe flow regime while the term "natural conditions" is used to refer to the river topography at a specific location.*

*Moving to the choice of the manuscript title (as well as our contribution), we would like to begin by noticing that reservoir operations could affect the model parameterization. Calibrating hydrological models with and without the representation of reservoirs could result in different sets of model parameters, even though they both have good model performances (i.e., comparable simulated and observed discharges). This is because, when reservoir operations are not included, model calibration adopts parameterizations that compensate for the absence of the reservoirs (Dang et al., 2020). In our case study, the discharge (observed or inferred from satellite data) includes the flow modification caused by reservoirs (see Section 4.1.3), so the reservoir operations must be included in the model. Specifically, we do that by integrating the reservoir operations estimated from satellite data (to replace the measured data due to their unavailability) into the hydrological model. To the best of our knowledge, this specific approach is still at its infancy in the hydrological modelling domain, so this is why we would like to retain the term "heavily regulated basins" in the title.*

*Finally, we note (and agree with the reviewer) that our approach is not only applicable to regulated basins. In fact, the reservoir operation module could be switched off when working on a natural catchment. We elaborated on this point in Section 5 (Lines 493-495 in the mark-up version of the revised manuscript).*

*References*

Bonnema, M. and Hossain, F.: Inferring reservoir operating patterns across the Mekong Basin using only space observations, Water Resources Research, 53, 3791-3810, https://doi.org/10.1002/2016wr019978, 2017.

Costa-Cabral, M. C., Richey, J. E., Goteti, G., Lettenmaier, D. P., Feldkötter, C., and Snidvongs, A.: Landscape structure and use, climate, and water movement in the Mekong River Basin, Hydrological Process, 22, 1731-1746, https://doi.org/10.1002/hyp.6740, 2007.

Dang, T. D., Chowdhury, A. F. M. K., and Galelli, S.: On the representation of water reservoir storage and operations in large-scale hydrological models: Implications on model parameterization and climate change impact assessments, Hydrology and Earth System Sciences, 24, 397–416, https://doi.org/10.5194/hess-24-397-2020, 2020.

Du, T. L. D., Lee, H., Dui, D. D., Arheimer, B., Li, H., Olsson, J., Darby, S. E., Sheffield, J., Kim, D., and Hwang, E.: Streamflow prediction in "geopolitically ungauged" basins using satellite observations and regionalization at subcontinental scale, Journal of Hydrology, 588, 125016, https://doi.org/10.1016/j.jhydrol.2020.125016, 2020.

Egüen, M., Aguilar, C., Herrero, J., Millares, A., and Polo, M. J.: On the influence of cell size in physically-based distributed hydrological modelling to assess extreme values in water resource planning, Natural Hazards Earth System Sciences, 12, 1573-1582, https://doi.org/10.5194/nhess-12-1573-2012, 2012.

Gupta, H. V., Kling, H., Yilmaz, K. K., and Martinez, G. F.: Decomposition of the mean squared error and NSE performance criteria: Implications for improving hydrological modelling, Journal of Hydrology, 377, 80-91, https://doi.org/10.1016/j.jhydrol.2009.08.003 2009.

Kabir, T., Pokhrel, Y., and Felfelani, F.: On the precipitation-induced uncertainties in process-based hydrological modeling in the Mekong River Basin, Water Resources Research, 58, e2021wr030 828, https://doi.org/10.1029/2021wr030828, 2022.

Tatsumi K. and Yamashiki, Y.: Effect of irrigation water withdrawals on water and energy balance in the Mekong River Basin using an improved VIC land surface model with fewer calibration parameters, Agricultural Water Management, 159, 92-106, http://doi.org/10.1016/j.agwat.2015.05.011, 2015.

Vu, D. T., Dang, T. D., Galelli, S., and Hossain, F.: Satellite observations reveal 13 years of reservoir filling strategies, operating rules, and hydrological alterations in the Upper Mekong River basin, Hydrology and Earth System Sciences, 26, 2345-2364, https://doi.org/10.5194/hess-26-2345-2022, 2022.

**Reply on RC2: 'Comment on hess-2023-35', Andrea Galletti,**

The authors propose an interesting framework for exploiting satellite data for the calibration of large-scale hydrological model. The steps undertaken to translate satellite-derived information into are explained in thorough detail. Then, the co-dependence between the hydrological model's performance and the calibration of the hydraulic model based on remote-sensed data is analyzed via Global Sensitivity Analysis. Finally, the authors adopt the satellite-derived information to calibrate VIC-Res and validate the predictions against available ground observation.

The concept is promising, however it would benefit from some considerations on its general applicability were drawn. Furthermore, the densely-flowing work sometimes overshadows the connection between the overall proposed framework and the particular tool/result being explained. I recommend acceptance of the paper, provided that the authors address or clarify the following points (minor revision):

*Response: Thank you for the positive feedback as well as the useful comments for improving the paper. In particular, we clarified a number of technical aspects (e.g., interpolation of monthly storage data, estimation of the river cross-section) and broadened the framing of our contribution / research questions. Additional details are reported below.*

1. The paper addresses the issue of calibrating macro-scale models in ungauged catchments. The introduction seems to implicitly assume that every large-scale hydrological model is always calibrated, and that this is most times done against available streamflow time series. However, several large scale hydrological models do not actually undergo case-specific calibration. While a calibrated model can provide results that are generally more reliable or realistic, this might not be clear to every reader, or someone could disagree. I recommend addressing the importance and effectiveness of calibration at the beginning (lines up to 15).

*Response: We agree with you. We made this point clear at the beginning of Introduction (Lines 18-19 in the mark-up version of the revised manuscript).*

2. The subsequent paragraph (line 17 onwards) could also benefit from a more thorough introduction to models' calibration and what is needed. In particular, it is not clearly explained why and how the presence of hydropower infrastructures for which operations are not known can create pitfalls for model calibration, nor why could it be at all important to include such factors in a calibration of hydrological model's parameters.

*Response: We agree with you. We addressed these points in the Introduction. Specifically, we (1) explained how the presence of hydraulic infrastructures (dams, in particular) affects model parameterization and (2) highlighted the importance of including such factor in the model calibration process (Lines 28-33 in the mark-up version of the revised manuscript).*

3. Some references and examples should be provided to support the sentence "Yet, this approach may still partially rely on in-situ data": this sentence opens up to the second class of RS-based works and basically justifies the study, therefore it needs to be backed by clear examples.

*Response: We revised it according to your suggestion by adding an example to support the sentence (Lines 50-53 in the mark-up version of the revised manuscript).*

4. Following point #1, I feel like a non-trivial question is whether it is possible and helpful to calibrate large scale hydrological models in ungauged catchments. This opens up to research questions I, II and III, helping to place them in a context that is broader than a mere numerical exercise.

*Response: Thank you for this comment, which pushed us to think more about the broader context of our work. As already pointed out in point #1, several large-scale hydrological models do not actually undergo case-specific calibration. We know that this can often lead to poor model performances. Hence, calibration has been suggested as a key avenue for model improvement---whenever computationally feasible (e. g., Bierkens et al., 2015; Samaniego et al., 2017). However, a prerequisite for a "successful" calibration is the availability of "good quality" data (i.e., at a minimum, "sufficiently long" time series of discharge observations and with "sufficiently small" observational errors). In many regions of the world, such good quality data are not available from in-situ observation networks, so the question arises whether remote-sensed data, despite all their uncertainty, are still "good enough" to support model calibration. Hence, the overarching question that this paper addresses is: to what extent is it possible and helpful to calibrate a macro-scale hydrological model in ungauged catchments using remote-sensed data? How do we deal with potential interactions between parameters used in data pre-processing (i.e., from remote-sensed data to reconstructed discharge data) and parameters of the hydrological models when doing model calibration? Can we reduce the uncertainty from such interactions in model calibration results? We made the overarching question underpinning our work clearly illustrated in the Introduction (see Lines 61-68 in the mark-up version of the revised manuscript).*

5. The choice of the 2009-2018 time window is not motivated as of line 84. I suspect the decision was taken backwards for compatibility with data coverage and quality (Figure 3). If this is the case, it should be stated clearly and the corresponding statements (line 96 and caption of fig 3) adjusted. If the choice was driven by something else, it should be pointed out.

*Response: Yes, the reason is not only to include the filling period of the two largest reservoirs (as stated in line 83-84 in the initial manuscript), but also to account for coverage and temporal resolution of altimetry data. We made these points clearer (Lines 110-112 in the mark-up version of the revised manuscript) and revised the corresponding statements (Lines 123-124, 128-129 in the mark-up version of the revised manuscript and the caption of Figure 3---which is the panel b of Figure 2 in the revised manuscript).*

6. I am dubious about the reliability of interpolating (linearly) daily reservoir operations between two monthly values. Hydropower often operates at the daily scale (or lower). The configuration of cascading dams might help masking errors in this procedure and their repercussions on the performance metrics. Did the authors compare their results with a fully-natural setup in order to evaluate the impacts of the reconstructed reservoir operation?

*Response: To answer this comment, let us first elaborate on how reservoirs in the Upper Mekong River have been designed and operated. Xiaowan and Nuozhadu are the two largest reservoirs: they have a massive capacity (~36 km³) and account for about 85 % of the total system's storage. Because of their size, their role is not to follow inter and intra-daily electricity demand variability, but rather to ensure a stable supply of power and to minimize the variability in the production of the other dams composing the hydropower system. This goal is reflected by their operating patterns. In the wet season (June-November), Xiaowan and Nuozhadu reservoirs gradually store water until reaching their maximum operational level (and release extra water if necessary). The other reservoirs run at their normal operational level (full capacity for power generation). In the dry season (December-May), Xiaowan and Nuozhadu gradually release water to the downstream reservoirs to ensure that the other reservoirs can run at their normal operational level. Putting this information together, we believe it is fair to state that Xiaowan and Nuozhadu are characterized by slow-varying dynamics (International Rivers, 2014; Vu et al., 2022). Hence, basing our analysis on interpolated monthly values is reasonable, although not ideal, of course. Our argument is partially supported by the analysis carried out in Vu et al. (2022), where we successfully compared the monthly storage of Xiaowan and Nuozhadu derived from Landsat images against the storage derived from Jason altimetry data (10-day temporal resolution) and Sentinel-1/2 images (6-day temporal resolution). Because of the spatial and temporal coverages of those data, we use the result derived from Landsat images for this study. We understand that all the information mentioned above is important. We added it to Section 3.1.2 (Lines 190-203 in the mark-up version of the revised manuscript).*

*A comparison between the river discharge simulated with and without reservoirs is technically feasible (see the figure below). However, we do not fully understand how this comparison can help evaluate the process of interpolating daily reservoir operations between two monthly values, and thus we believe that the explanation provided above is more compelling. Hence, we suggest including this comparison in the response to reviewers but not in the revised manuscript.*

[Figure]

*Comparisons between simulated discharge with dams (dark green) and without dams (yellow green) at the virtual station (a) and Chiang Saen station (b). The variation ranges of the two settings (with and without dams) are corresponding to the twelve selected solutions in calibration (explained in Section 4.3).*

7. The estimation of the river cross section represents an important step in the framework, influencing the outcomes of the rating curve and all subsequent analyses. With the river width ranging from 400m to 200m depending on water level, and with DEM and Landsat images having a resolution of 30m, one could think that the combination of cells taken as the cross section is not univoque. Exploring multiple alternatives for the chosen cross section could help assessing the variability of this extrapolation (if at all present). Otherwise, a plane view or schematic of the chosen pixels would help understanding the decision taken. Furthermore, river bed might be subjected to vertical evolution (i.e., erosion or sediment accumulation) which is hardly grasped by this methodology since it relies on a static DEM. Performing a similar extrapolation (of the river cross section) e.g.,at location 812 which is closer to Chiang Saen and then comparing the so-obtained discharge values with ground observations would have provided a rather solid, yet not exhaustive, base for any assumption made at Virtual Station, reinforcing the implicit validation described at lines 267-269.

*Response: Thank you for the suggestion of exploring multiple alternatives for the chosen cross-section. In the panels a-d of the figure below, we show the results of four alternative cross-sections, created by moving the one at the location of the virtual station (reported in the manuscript, marked by VS in the panel e and f) 30 and 60m (1 and 2 cells) both upstream and downstream. The alternatives are well in agreement with the one reported in the manuscript. Specifically, riverbed elevations are 277.2, 275.6, 276, 274.5, and 274.3 m a.s.l. (from upstream to downstream). We plot the five cross-sections together in panel e, and show a 3D visualization in panel f. We added this point to Section 4.1.1 (Lines 304-309 in the mark-up version of the revised manuscript) and added the plot below to the Supplement (Figure S2).*

*We agree with the second point that riverbeds might be subjected to vertical evolution, which is hardly grasped by a static DEM. We added this point as one of the limitations of the methodology in Section 5 (Lines 468-469 in the mark-up version of the revised manuscript).*

*As for the last point, we believe that applying our methodology to location 812 and other locations which are close to Chiang Saen station may not be possible. This is due to a number of issues: (1) the temporal resolution (35 and 27 days for Envisat and Sentinel-3A respectively) and coverage (see Figure 3) of altimetry data is too coarse; (2) Manning's equation works best for straight river segments with limited discharge variations due to tributaries and distributaries nearby (Przedwojski et al., 1995), while most locations close to Chiang Saen are located in curved river segments, where many tributaries join the mainstream of the Mekong River; and (3) modifications of the riverbank topography (for road and agriculture/farming) around most of those locations affect river cross-section estimations. In the mark-up version of the revised manuscript, we elaborated on these points in Section 2.3 (Lines 128-129), Section 3.2 (Lines 221-224, 236-240), and Section 5 (Lines 484-492), where we expanded the discussion on the limitations of our methodology.*

[Figure]

*Figure S2: Panels (a-d) show the results of four alternative cross-sections, created by moving the one at the location of the virtual station (reported in Figure 4a in the revised manuscript and marked by VS in the panel (e) and (f)) 30 and 60m (1 and 2 cells) both upstream and downstream. The five cross-sections are plotted together in panel (e), and a 3D visualization is provided in panel (f).*

8. Line 338 onwards: the selected 25% is said to be the best according to all metrics: does this mean that the (i assume normalized, as defined in the methods) metrics were averaged and the best 25% averages were taken? This concept could be made a bit clearer. Furthermore, it is unclear what determined the choice of the 12 and 58 stations presented later in this section.

*Response: To select the solutions we first created four groups (each containing the top 25% solutions w.r.t. a single performance metric) and then took their intersection. This approach yields 12 solutions. The other approach we considered is the definition of Pareto efficiency, which yields 58 solutions. We made this point clear to avoid any misunderstandings (Lines 409-412, 426-428 in the mark-up version of the revised manuscript).*

9. Line 359: Phrasing here is a bit misleading, and a more thorough introduction could be useful: GSA was used to determine the co-dependence between n and the model's performance, while the potential bias intrinsic to the co-dependence was not really assessed, although one could reasonably suppose that it is present. Providing further evidence (references) that joint estimation of discharge and parameters leads to a biased calibration could help strenghten the need to break the co-dependence.

*Response: Yes, we agree with you that this part is potentially misleading, so we revised it accordingly. Also, thank you for your suggestion of providing further evidence (references) that joint estimation of discharge and parameters leads to a biased calibration. We note that there is a very limited number of studies that look specifically at the problem of biased calibration (when co-calibrating a hydraulic and hydrological model). We added a reference (Lima et al., 2019) to strengthen our related statements in Section 1 (Lines 54-56 in the mark-up version of the revised manuscript) and Section 5 (Lines 442-444 in the mark-up version of the revised manuscript).*

TECHNICAL CORRECTIONS:

Line 227: is 2008-2018 correct? it seems it should be 2009
*Response: Yes. You are right. It is 2009-2018 (Line 283 in the mark-up version of the revised manuscript).*

Line 297: the value in parentheses should be 0.045
*Response: Yes. You are right. It is 0.045 (Line 363 in the mark-up version of the revised manuscript).*

Line 382-383: do not contribute to (reducing) modelling uncertainty?
*Response: We revised the sentence (Line 467 in the mark-up version of the revised manuscript).*

Line 393: One often recurring... one should be removed?
*Response: We improved the readability of this sentence (Lines 497 in the mark-up version of the revised manuscript).*

*References*

Bierkens, M. F. P.: Global hydrology 2015: State, trends, and directions, Water Resources Research, 51, 4923-4947, https://doi.org/10.1002/2015wr017173, 2015.

International Rivers: The Environmental and Social Impacts of Lancang Dams, https://archive.internationalrivers.org/sites/default/files/attached-files/ir_lancang_dams_researchbrief_final.pdf, last access 11 April 2023, 2014.

Lima, F. N., Fernandes, W., and Nascimento, N.: Joint calibration of a hydrological model and rating curve parameters for simulation of flash flood in urban areas, Brazilian Journal of Water Resources, 24, https://doi.org/10.1590/2318-0331.241920180066, 2019.

Przedwojski, B., Blazejewski, R., Pilarczyk, K.: River training techniques: Fundamentals, design and applications, Netherlands: Taylor & Francis, 1995.

Samaniego, L., Kumar, R., Thober, S., Rakovec, O., Zink, M., Wanders, N., Eisner, S., Schmied, H. M., Sutanudjaja, E. H., Warrach-Sagi, K., and Attinger, S.: Toward seamless hydrologic predictions across spatial scales, Hydrology Earth System Sciences, 21, 4323-4346, https://doi.org/10.5194/hess-21-4323-2017, 2017.

Vu, D. T., Dang, T. D., Galelli, S., and Hossain, F.: Satellite observations reveal 13 years of reservoir filling strategies, operating rules, and hydrological alterations in the Upper Mekong River basin, Hydrology and Earth System Sciences, 26, 2345-2364, https://doi.org/10.5194/hess-26-2345-2022, 2022.